# Expanded genetic screening in *Caenorhabditis elegans* identifies new regulators and an inhibitory role for NAD+ in axon regeneration

Kyung Won Kim[1‡*], Ngang Heok Tang[1], Christopher A Piggott[1†], Matthew G Andrusiak[1†], Seungmee Park[1†], Ming Zhu[1], Naina Kurup[1], Salvatore J Cherra III[1§], Zilu Wu[1], Andrew D Chisholm[1*], Yishi Jin[1,2*]

[1]Section of Neurobiology, Division of Biological Sciences, University of California, San Diego, La Jolla, United States; [2]Department of Cellular and Molecular Medicine, University of California, San Diego, School of Medicine, La Jolla, United States

*For correspondence:
kwkim@hallym.ac.kr (KWK);
adchisholm@ucsd.edu (ADC);
yijin@ucsd.edu (YJ)

[†]These authors contributed equally to this work

Present address: [‡]Convergence Program of Material Science for Medicine and Pharmaceutics, Department of Life Science, Multidisciplinary Genome Institute, Hallym University, Chuncheon, Republic of Korea; [§]Department of Neuroscience, University of Kentucky College of Medicine, Lexington, United States

Competing interests: The authors declare that no competing interests exist.

**Abstract** The mechanisms underlying axon regeneration in mature neurons are relevant to the understanding of normal nervous system maintenance and for developing therapeutic strategies for injury. Here, we report novel pathways in axon regeneration, identified by extending our previous function-based screen using the *C. elegans* mechanosensory neuron axotomy model. We identify an unexpected role of the nicotinamide adenine dinucleotide (NAD+) synthesizing enzyme, NMAT-2/NMNAT, in axon regeneration. NMAT-2 inhibits axon regrowth via cell-autonomous and non-autonomous mechanisms. NMAT-2 enzymatic activity is required to repress regrowth. Further, we find differential requirements for proteins in membrane contact site, components and regulators of the extracellular matrix, membrane trafficking, microtubule and actin cytoskeleton, the conserved Kelch-domain protein IVNS-1, and the orphan transporter MFSD-6 in axon regrowth. Identification of these new pathways expands our understanding of the molecular basis of axonal injury response and regeneration.
DOI: https://doi.org/10.7554/eLife.39756.001

## Introduction

Axon regeneration after injury is an important and conserved biological process in many animals, involving a large number of genes and pathways (*He and Jin, 2016*; *Mahar and Cavalli, 2018*; *Tedeschi and Bradke, 2017*). Upon axonal injury, distal axon segments degenerate and segments proximal to the cell body remain alive and can in certain cases regenerate (*Chen et al., 2007*; *McQuarrie and Grafstein, 1973*; *Neumann and Woolf, 1999*). Axon regeneration after injury requires rapid sealing of the damaged plasma membrane (PM) and subsequent formation of growth cones, leading to regrowth and extension from damaged proximal axons. These cellular changes involve numerous molecular pathways, starting with rapid calcium influx at injury sites (*Ghosh-Roy et al., 2010*; *Rishal and Fainzilber, 2014*; *Wolf et al., 2001*), retrograde injury signaling, transcriptional reprogramming to re-structuring of the cytoskeleton and re-organization of the extracellular matrix (ECM) (*Blanquie and Bradke, 2018*). In the adult mammalian central nervous system (CNS), axon regeneration is limited, due to the combination of a repressive glial environment and a lower intrinsic growth capacity of CNS neurons (*He and Jin, 2016*). The lack of axonal regrowth after CNS injuries, therefore, impairs functional recovery.

Many approaches have been proposed and tested to promote axon regeneration over the past decades (*David and Aguayo, 1981*; *He and Jin, 2016*; *Park et al., 2008*). Yet, mechanistic

understanding of how damaged axons regenerate in a permissive environment remains fragmented. Since the discovery of functional axon regeneration in the nematode *Caenorhabditis elegans* (*Yanik et al., 2004*), several function-based genetic screens have revealed conserved axon regeneration genes and pathways, notably the highly conserved MAPKKK DLK-1 signaling cascade (*Yan et al., 2009*; *Chen et al., 2011*; *Hammarlund et al., 2009*; *Nix et al., 2014*). We previously reported a distinct set of genes identified from a genetic screen of 654 genes in mechanosensory axon regeneration (*Chen et al., 2011*). For example, regulators of microtubule (MT) dynamics play a rate-limiting role in axon regrowth, consistent with findings from other animal models (*Bradke et al., 2012*; *Hur et al., 2012*). Additional studies reveal other conserved pathways include the RNA-binding protein CELF/UNC-75 (*Chen et al., 2016a*), the miRNA and piRNA pathway (*Kim et al., 2018*; *Zou et al., 2013*), the fusogen EFF-1 (*Ghosh-Roy et al., 2010*; *Neumann et al., 2015*), and the apoptotic pathway (*Pinan-Lucarre et al., 2012*). Importantly, the findings from *C. elegans* are echoed from similar screening in mammalian neurons (*Sekine et al., 2018*; *Zou et al., 2015*).

Here, we report our analysis of 613 additional new genes using the *C. elegans* mechanosensory axon regeneration assay. We find new gene classes with inhibitory roles in axon regrowth, such as the NAD$^+$ salvage pathway and the conserved Kelch-domain protein IVNS-1. We also find several permissive factors, such as A Disintegrin and Metalloprotease with Thrombospondin repeats (ADAMTS) proteins, a Rab GTPase RAB-8, and the membrane transporter MFSD-6. We show that the endoplasmic reticulum (ER)-PM contact site protein Extended Synaptotagmin (ESYT-2) is sensitive to axonal injury, and that Junctophilin (JPH-1) inhibits axon-axon fusion. Our studies of genes encoding lipid or phospholipid metabolic enzymes indicate extensive functional redundancy. This expanded screen reinforces several themes from the previous study, such as the inhibitory role of ECM components and the permissive role of MT stabilization (*Chen et al., 2011*). Together, our findings highlight the molecular complexity of axon regeneration and provide the genetic framework for a more comprehensive understanding of axon regeneration.

## Results

We screened 613 additional genes representing nine classes of protein function and structure, selected based on their sequence conservation and the availability of viable genetic mutants with normal axon development (*Figure 1A,B*; *Figure 1—source data 1*). We tested genetic null or strong loss-of-function mutations in each gene for effects on mechanosensory PLM (Posterior Lateral Microtubule) axon regeneration. In the PLM axon regrowth model, we sever the axon ~50 µm distal from the cell body in the fourth larval (L4) stage using a femtosecond laser and measure axon regrowth 24 hr post-axotomy in at least 10 animals per strain (*Wu et al., 2007*). From these 613 genes, we identified 49 genes promoting PLM regrowth (i.e. showing reduced regrowth in loss-of-function mutants) and 34 genes inhibiting regrowth (i.e. increased regrowth in loss-of-function mutants) (*Tables 1* and *2*; *Figure 1—source data 1*). As in our previous screen, genes affecting axon regrowth are found across all functional and structural classes tested (*Figure 1B*). The percentage of genes having positive or negative effects on regrowth was similar to that reported in our previous screen (*Chen et al., 2011*) (*Figure 1—figure supplement 1*), suggesting this screen remains far from saturated. The combined analyses of >1200 genes reinforce the conclusion that regenerative axon regrowth requires many genetic pathways, most of which are not involved in developmental axon outgrowth or guidance. Below, we first focus on a set of genes with previously uncharacterized roles in axon regeneration and then summarize common themes from the expanded screen.

### The conserved enzyme NMNAT inhibits axon regeneration

Among genes with significant inhibitory effects on axon regrowth, we identified NMAT-2, a member of the nicotinamide mononucleotide adenylyltransferase (NMNAT) enzyme family (*Figure 2A,B*). NMNAT enzymes catalyze a vital step in NAD$^+$ biosynthesis and confer neuroprotection in several injury models of flies and mice (*Gerdts et al., 2016*). In mammalian neurons increasing NMNAT activity protects against Wallerian degeneration and axon degradation following trophic factor withdrawal (*Mack et al., 2001*; *Vohra et al., 2010*). In *C. elegans*, overexpression of NMAT-2/ NMNAT protects against neuronal degeneration caused by the toxic mutant ion channel MEC-4(d) (*Calixto et al., 2012*), but does not protect against distal axon degeneration after laser axotomy (*Nichols et al., 2016*). We found that PLM regrowth was enhanced in two independent *nmat-2* null

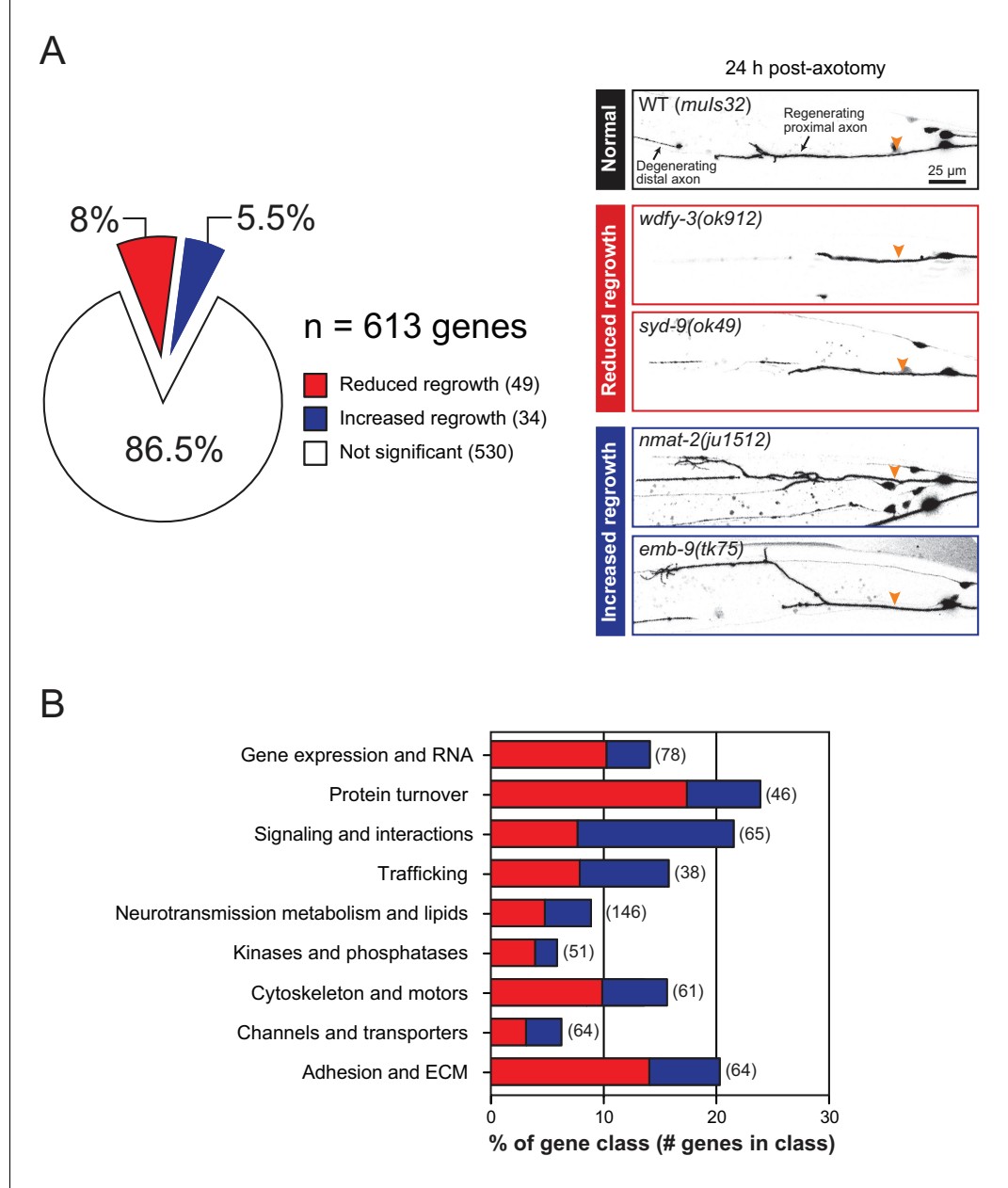

**Figure 1.** Overview and results of expanded axon regrowth screen. (**A**) Pie chart showing fraction of genes screened displaying significantly reduced or increased regrowth at 24 hr. Right: representative inverted grayscale images of PLM 24 hr post-axotomy in wild type (black box), and mutants with reduced (red boxes) or increased regrowth (blue boxes). Orange arrowhead, site of axotomy. (**B**) Distribution of reduced/increased regrowth mutants among nine functional or structural gene classes, shown as percentage of genes in each class. See *Figure 1—source data 1* for lists of genes in each class.

DOI: https://doi.org/10.7554/eLife.39756.002

The following source data and figure supplements are available for figure 1:

**Source data 1.** List of screened genes, reference alleles, and the functional categories.
DOI: https://doi.org/10.7554/eLife.39756.006

**Figure supplement 1.** The overview and results of axon regrowth screen combined with our previous study.
DOI: https://doi.org/10.7554/eLife.39756.003

**Figure supplement 2.** Mutants affecting multiple biological processes required for normal axon regrowth.
DOI: https://doi.org/10.7554/eLife.39756.004

**Figure supplement 2—source data 1.** Each data point in all graphs.
DOI: https://doi.org/10.7554/eLife.39756.005

**Table 1.** Mutants displaying reduced PLM regrowth

| Gene name | Mutations | Normalized regrowth (24 hr) | N | P value | Molecular function | Closest human Gene[a] |
|---|---|---|---|---|---|---|
| A. Cell Adhesion and ECM | | | | | | |
| adt-1 | cn30 | 0.75 | 21 | ** | ADAM metalloprotease | ADAMTS3 |
| adt-3 | ok923 | 0.71 | 46 | ** | ADAM metalloprotease | ADAMTS2 |
| C05D9.7 | ok2931 | 0.60 | 28 | *** | Unknown | N/A |
| dpy-10 | e128 | 0.67 | 28 | ** | 5FMC ribosome biogenesis complex | PELP1 |
| F35G2.1 | ok1669 | 0.68 | 27 | *** | Quiescin sulfhydryl oxidase | QSOX1 |
| gly-2 | tm839 | 0.69 | 16 | * | Mannosyl-glycoprotein N-acetylglucosaminyltransferases | MGAT5 |
| osm-11 | rt142 | 0.72 | 13 | ** | Secreted protein | N/A |
| zig-1 | ok784 | 0.68 | 12 | * | Basigin | BSG |
| zig-3 | ok1476 | 0.73 | 18 | ** | Kazal type serine peptidase inhibitor domain | KAZALD1 |
| B. Channels and transporters | | | | | | |
| abts-1 | ok1566 | 0.74 | 34 | *** | Anion exchange protein | SLC4A7 |
| mfsd-6 | ju833 | 0.68 | 15 | ** | Major facilitator | MFSD6 |
| C. Cytoskeleton and motors | | | | | | |
| fli-1 | ky535 | 0.61 | 27 | *** | Actin remodeling protein | FLII |
| mec-12 | e1605 | 0.73 | 15 | * | Tubulin α−3 chain | TUBA1C |
| mec-17 | ok2109 | 0.55 | 14 | ** | α-Tubulin N-acetyltransferase | ATAT1 |
| tba-9 | ok1858 | 0.70 | 24 | ** | α-Tubulin | TUBA3 |
| vab-10 | e698 | 0.78 | 11 | * | Spectraplakin | DST |
| D. Protein kinases and phosphatases | | | | | | |
| plk-1 | or683ts | 0.59 | 13 | ** | Polo like kinase | PLK1 |
| svh-2 | tm737 | 0.68 | 30 | *** | Receptor Tyrosine kinase | MET |
| E. Neurotransmission, metabolism, and lipid | | | | | | |
| cept-2 | ok3135 | 0.68 | 27 | * | Choline/ethanolamine phosphotransferase | CEPT1 |
| cpr-1 | ok1344 | 0.60 | 32 | *** | Cysteine proteinase | CTSB |
| dhhc-11 | gk1105 | 0.75 | 32 | *** | Palmitoyltransferase | ZDHHC11 |
| eat-3 | tm1107 | 0.74 | 16 | ** | Mitochondrial dynamin like GTPase | OPA1 |
| npr-20 | ok2575 | 0.49 | 44 | *** | G-protein coupled receptor | CCKBR/TRHR |
| ptps-1 | tm1984 | 0.55 | 27 | *** | 6-pyruvoyl tetrahydrobiopterin synthase | PTS |
| supr-1 | ju1118 | 0.78 | 30 | ** | Unknown | N/A |
| F. Trafficking | | | | | | |
| jph-1 | ok2823 | 0.77 | 14 | ** | Junctophilin | JPH1 |
| rab-8 | tm2526 | 0.77 | 28 | ** | Ras GTPase | RAB8B |
| rsef-1 | ok1356 | 0.66 | 17 | * | Endosomal Rab family GTPase | RASEF |
| G. Signaling and interactions | | | | | | |
| osm-7 | tm2256 | 0.57 | 28 | ** | Unknown | N/A |
| par-2 | or373 | 0.63 | 47 | *** | C3HC4-type RING-finger | TRIM |
| rgl-1 | ok1921 | 0.74 | 30 | ** | Ral guanine nucleotide dissociation stimulator | RGL1 |
| wdfy-3 | ok912 | 0.52 | 17 | *** | WD40 and FYVE domain | WDFY3 |
| wdr-23 | tm1817 | 0.66 | 30 | *** | DDB1 and CUL4 associated factor | DCAF11 |
| H. Protein turnover, proteases, cell death | | | | | | |
| brap-2 | ok1492 | 0.61 | 62 | *** | BRCA1-associated protein; zinc ion binding activity | BRAP |
| cdc-48.1 | tm544 | 0.69 | 19 | ** | Transitional ER ATPase homolog | VCP |
| ced-9 | n1950 | 0.73 | 16 | * | Cell-death inhibitor Bcl-2 homolog | BCL2 |

*Table 1 continued on next page*

*Table 1 continued*

| Gene name | Mutations | Normalized regrowth (24 hr) | N | P value | Molecular function | Closest human Gene[a] |
|---|---|---|---|---|---|---|
| dnj-23 | tm7102 | 0.69 | 32 | ** | DNaJ domain (prokaryotic heat shock protein) | DNAJC9 |
| fbxc-50 | tm5154 | 0.73 | 12 | * | F-box protein | N/A |
| math-33 | ok2974 | 0.64 | 12 | *** | Ubiquitin-specific protease | USP7 |
| skr-5 | ok3068 | 0.69 | 12 | * | S-phase kinase associated protein | SKP1 |
| tep-1 | tm3720 | 0.53 | 36 | *** | ThiolEster containing Protein; endopeptidase inhibitor activity | CD109 |
| **I. Gene expression and RNA regulation** | | | | | | |
| mec-8 | e398 | 0.23 | 16 | *** | RNA binding protein, mRNA processing factor | RBPMS |
| rict-1 | mg360 | 0.58 | 26 | *** | Subunit of TORC2 | RICTOR |
| rtcb-1 [b] | gk451 | 0.58 | 25 | *** | tRNA-splicing ligase RtcB homolog | RTCB |
| skn-1 | ok2315 | 0.78 | 10 | * | Basic leucine zipper protein | NFE2 |
| smg-3 | r930 | 0.68 | 28 | *** | Nonsense mediated mRNA decay regulator | UPF2 |
| syd-9 | ju49 | 0.47 | 15 | *** | Zinc finger E-box binding homeobox | ZEB1 |
| tdp-1 | ok803 | 0.70 | 36 | *** | TAR DNA-binding protein | TARDBP/TDP-43 |
| wdr-5.1 | ok1417 | 0.70 | 26 | *** | WD repeat-containing protein | WDR5 |

Genes are classified in nine functional or structural classes. Mutations are genetic or predicted molecular nulls, or partial loss-of-function. Normalized regrowth is relative to matched same-day controls or to pooled controls. Significant levels (*p<0.05; **p<0.01; ***p<0.001) based on Student's *t*-test.
[a] Closest human gene based on BLASTP score in Wormbase WS263; Ensembl/HGNC symbol.
[b] *rtcb-1(gk451)* mutant reported to show increased regrowth in the *C. elegans* motor neurons (**Kosmaczewski et al., 2015**).
DOI: https://doi.org/10.7554/eLife.39756.007

(*0*) mutants, *tm2905* and *ju1512* (**Figure 2B,C**). A null mutation of NMAT-1, a close paralog, did not affect PLM regrowth (**Figure 2C**). *nmat-2(0)* adult animals are sterile, while *nmat-1(0)* are fertile, indicating that these two NMNATs may have distinct tissue- or cell-type-specific roles.

To address whether the observed effects of *nmat-2(0)* are related to $NAD^+$ synthesis, we examined loss-of-function mutants of other enzymes in the invertebrate $NAD^+$ salvage synthesis pathway (**Figure 2A**), including the glutamine-dependent $NAD^+$ synthase QNS-1, nicotinamide riboside kinase (NRK) NMRK-1, nicotinate phosphoribosyltransferase (NAPRT) NPRT-1, nicotinamidase PNC-1 and PNC-2 (**Magni et al., 1999**; **Vrablik et al., 2009**). Among these, only *qns-1(0)* mutants showed marginally increased axon regrowth (**Figure 2C**). NMAT-2 and QNS-1 catalyze the terminal steps of the $NAD^+$ salvage pathway. Like *nmat-2(0)*, *qns-1(0)* mutants are sterile (**Wang et al., 2015**) (this work), while other single mutants are fertile, suggesting that NMAT-2 and QNS-1 define essential steps in the biosynthesis of $NAD^+$. To address whether sterility of the animals might contribute to the observed effects on axon regrowth, we cultured animals on 5'fluoro-2' deoxyuridine (FUdR) and found that neither wild type or *nmat-1(0)* grown in FUdR showed increased PLM regrowth (**Figure 2—figure supplement 1**). Additionally, we have previously reported that sterile animals following germline ablation do not affect PLM regrowth (**Kim et al., 2018**). Thus, we conclude that NMAT-2's role in axon regrowth is independent of animal fertility.

We next focused on NMAT-2 to define the role of $NAD^+$ pathway in axon regeneration. Using CRISPR genome editing, we generated a single copy transgene expressing *nmat-2(+)* under its endogenous promoter (*juSi347*). This transgene fully rescued the sterility of *nmat-2(0)* and restored the increased axon regrowth in *nmat-2(0)* mutants to wild-type levels (**Figure 2C**), confirming that the increased axon regrowth is due to loss of NMAT-2 function. We then asked in which tissues NMAT-2 acts to inhibit axon regeneration using transgenic expression of NMAT-2 in the epidermis, intestine, or mechanosensory neurons (**Figure 2—source data 1**). Transgenic expression of NMAT-2 in individual tissues was not able to restore axon regeneration in *nmat-2(0)* to normal (**Figure 2D**). Interestingly, the combined expression of NMAT-2 in all three tissues restored normal axon regeneration (**Figure 2D**), and also partially rescued sterility. We conclude that NMAT-2 may act in both neuronal and non-neuronal cells to inhibit axon regeneration.

**Table 2.** Mutants displaying increased PLM regrowth

| Gene name | Mutations | Normalized regrowth (24 hr) | N | P value | Molecular function | Closest human Gene[a] |
|---|---|---|---|---|---|---|
| A. Cell adhesion and ECM | | | | | | |
| emb-9 | tk75 [b] | 1.37 | 28 | *** | Collagen type IV α3 chain | COL4A3 |
| epi-1 | gm121 | 1.33 | 36 | *** | Laminin | LAMA |
| mig-17 | k174 | 1.24 | 37 | *** | ADAM metalloprotease | ADAMTS5 |
| ZC116.3 | ok1618 | 1.40 | 26 | *** | Cubilin | CUBN |
| B. Channels and transporters | | | | | | |
| lgc-12 | ok3546 | 1.33 | 26 | ** | Serotonin receptor 3E | HTR3E |
| tmc-1 | ok1859 | 1.31 | 30 | ** | Transmembrane channel-like protein | TMC1 |
| C. Cytoskeleton and motors | | | | | | |
| ivns-1 | ok3171 | 1.31 | 18 | ** | Actin-binding; splicing | IVNS1ABP |
| twf-2 | ok3564 | 1.33 | 39 | ** | Twinfilin actin binding protein | TWF |
| nud-1 | ok552 | 1.30 | 25 | ** | Nuclear distribution C, Dynein complex regulator | NUDC |
| tba-7 | gk787939 | 1.45 | 15 | *** | α-tubulin | TUBA |
| E. Neurotransmission, metabolism, and lipid | | | | | | |
| nmat-2 | tm2905 | 1.55 | 38 | *** | Nicotinamide mononucleotide adenylyltransferase | NMNAT1 |
| qns-1 | ju1563 | 1.13 | 40 | * | NAD + synthetase | NADSYN1 |
| mgl-1 | tm1811 | 1.24 | 28 | ** | Glutamate metabotropic receptor | GRM3 |
| mgl-3 | tm1766 | 1.32 | 22 | ** | Glutamate metabotropic receptor | GRM6 |
| npr-25 | ok2008 | 1.27 | 26 | ** | Coagulation factor II thrombin receptor | F2RL2 |
| ucr-2.3 | ok3073 | 1.41 | 24 | *** | Ubiquinol-cytochrome C reductase core protein | UQCRC2 |
| F. Trafficking | | | | | | |
| nex-1 [c] | gk148 | 1.38 | 27 | *** | Annexin | ANXA13 |
| nex-2 [d] | ok764 | 1.23 | 30 | ** | Annexin | ANXA7 |
| snb-6 | tm5195 | 1.30 | 38 | ** | Vesicle associated membrane protein | VAMP1 |
| G. Signaling and interactions | | | | | | |
| drag-1 | tm3773 | 1.52 | 26 | ** | Repulsive guidance molecule BMP co-receptor | RGMB |
| ect-2 | ku427 | 1.31 | 31 | ** | RhoGEF | ECT2 |
| lin-2 | e1309 | 1.31 | 27 | *** | Membrane associated guanylate kinase | CASK |
| magi-1 | zh66 | 1.40 | 29 | *** | Membrane associated guanylate kinase | MAGI2 |
| prmt-5 | gk357 | 1.26 | 24 | ** | Protein arginine N-methyltransferase | PRMT5 |
| rap-1 | tm861 | 1.33 | 10 | *** | Ras small GTPase | RAP1 |
| smz-1 | ok3576 | 1.39 | 13 | * | PDZ domain-containing protein | N/A |
| trxr-1 | tm2047 | 1.3 | 31 | * | Thioredoxin reductase | TXNRD2 |
| H. Protein turnover, proteases, cell death | | | | | | |
| natb-1 | ju1405 | 1.29 | 14 | * | N(α)-acetyltransferase 20 | NAA20 |
| rnf-5 | tm794 | 1.28 | 15 | * | Ring finger protein | RNF5 |
| ulp-5 | tm3063 | 1.22 | 30 | *** | SUMO specific peptidase | SENP7 |
| I. Gene expression and RNA regulation | | | | | | |
| csr-1 | fj54 | 1.24 | 38 | *** | Argonaute | AGO1 |
| hda-6 | tm3436 | 1.29 | 42 | *** | Histone deacetylase | HDAC6 |
| elpc-3 | ok2452 | 1.31 | 26 | *** | Elongator acetyltransferase complex subunit | ELP3 |

Genes are classified in nine functional or structural classes. Mutations are genetic or predicted molecular nulls, or partial loss-of-function. Normalized regrowth is relative to matched same-day controls or to pooled controls. Significant levels (*p<0.05; **p<0.01; ***p<0.001) based on Student's *t*-test.

[a] Closest human gene based on BLASTP score in Wormbase WS263; Ensembl/HGNC symbol.

[b] *emb-9*(*tk75*) mutant reported to be a gain-of-function allele that makes stable EMB-9/Type IV collagen (**Kubota et al., 2012**).

[c] *nex-1*(*gk148*) mutant reported to show reduced regrowth in the *C. elegans* motor neurons (**Nix et al., 2014**).

<sup>d</sup> *nex-2*(*bas4*) mutant reported to show normal regrowth in the *C. elegans* motor neurons (**Nix et al., 2014**).
DOI: https://doi.org/10.7554/eLife.39756.008

In addition to their enzymatic roles, several NMNAT proteins function as molecular chaperones, including *Drosophila* NMNAT, mouse NMNAT2, and human NMNAT3 (**Ali et al., 2016**; **Zhai et al., 2006**; **Zhai et al., 2008**). We therefore tested whether the enzymatic properties of NMAT-2 are required for inhibition of axon regeneration. Using CRISPR genome editing, we mutated the active site motif involved in ATP recognition (**Zhang et al., 2002**) (**Figure 2B**). This mutant *nmat-2(ju1514)* displayed sterility and enhanced regrowth of PLM neurons (**Figure 2C**), indistinguishable from *nmat-2(0)* mutants. Therefore, the role of NMAT-2 in axon regeneration likely requires its enzymatic activity. Here, we infer that the enhanced axon regeneration in *nmat-2(0)* reflects sustained low levels of NAD$^+$.

The neuroprotective effect of NMNAT is cell-autonomous in *Drosophila* and in mice (**Gilley et al., 2013**; **Wen et al., 2011**). Our finding that NMAT-2 inhibits axon regrowth via several tissues suggests that NMNAT may function via distinct mechanisms for neuroprotection vs. axon regeneration. The PLM axon is adjacent to the intestine and is enveloped by the surrounding epidermis (**Emtage et al., 2004**). Speculatively, NAD$^+$ might activate inhibitory factors in neurons and in surrounding tissues, which act together to repress the axon regenerative response; some of these factors might regulate cell-cell interaction and signal transduction. In *Drosophila*, lack of NMNAT also led to enhanced sensory axon regeneration (**Chen et al., 2016b**). Together, these data suggest conserved roles of NMNAT in axon regeneration. Future work will be required to dissect specific mechanisms by which NMNAT inhibits axon regeneration.

## Differential roles and functional redundancy of ER-PM contact site components in axon regeneration

Membrane contact sites (MCSs) are regions where membranes from two organelles or an organelle and the PM are held together by protein tethers, most of which are conserved from yeast to mammals (**Phillips and Voeltz, 2016**; **Saheki et al., 2016**). MCSs can coordinate activities such as calcium entry or lipid transfer between membranes. Calcium entry via voltage-gated Ca$^{2+}$ channels in the PM is critical for PLM axon regeneration (**Ghosh-Roy et al., 2010**). Additionally, MCSs between the PM and ER might be involved in lipid addition to the PM during rapid extension of regrowing axons (**Hausott and Klimaschewski, 2016**). We therefore examined mutants affecting conserved ER-PM MCS components such as Junctophilin, Extended synaptotagmin (E-Syt), Anoctamins, and OxySterol Binding Proteins (OSBP).

Junctophilins are multi-pass transmembrane proteins that are localized to ER-PM contacts in excitable cells, where they couple PM- and ER-localized calcium channels (**Landstrom et al., 2014**). JPH-1 is the sole Junctophilin in *C. elegans* (**Yoshida et al., 2001**) (**Figure 3A**). We observed that *jph-1(ok2823)* mutants, likely null, exhibited a significantly increased rate of reconnection or fusion between the regrowing axon and distal fragment (**Figure 3B**). Axons that did not reconnect in *jph-1* mutants exhibited reduced axon regeneration, compared to controls (**Figure 3C**). As reconnected axons were not measured for regrowth analysis, the reduced regrowth in *jph-1* mutants might be due to an overrepresentation of poorly growing axons. Axon-axon fusion requires the fusogen EFF-1 (**Ghosh-Roy et al., 2010**; **Pérez-Vargas et al., 2014**) and a phosphoserine-mediated apoptotic cell engulfment pathway (**Neumann et al., 2015**). We analyzed *eff-1; jph-1* double mutants and found that the enhanced reconnection in *jph-1* was greatly reduced (**Figure 3B**). *Drosophila* Junctophilin-like molecule functions in apoptotic cell removal (**Gronski et al., 2009**). These observations suggest JPH-1-mediated contacts may restrict axon-axon fusion, dependent on *eff-1*.

Extended synaptotagmins (E-Syt) are a family of proteins containing multiple C2 domains (**Figure 3D**) that have been shown to tether the ER to the PM (**Giordano et al., 2013**) and are implicated in membrane lipid transfer (**Saheki et al., 2016**; **Yu et al., 2016**). ESYT-2 is the sole E-Syt in *C. elegans* and is most closely related to human E-Syt2 and E-Syt3 (**Figure 3D**). We found that *esyt-2* showed wide expression in the nervous system (**Figure 3—figure supplement 1**). In the mechanosensory neuron cell body, full-length GFP-ESYT-2 showed a punctate pattern, colocalizing with an ER marker PISY-1 (**Rolls et al., 2002**) at the peripheral ER (**Figure 3E**). In uninjured axons, ESYT-2 was distributed intermittently (**Figure 3F**; upper panel). Strikingly, upon axon injury, axonal ESYT-2

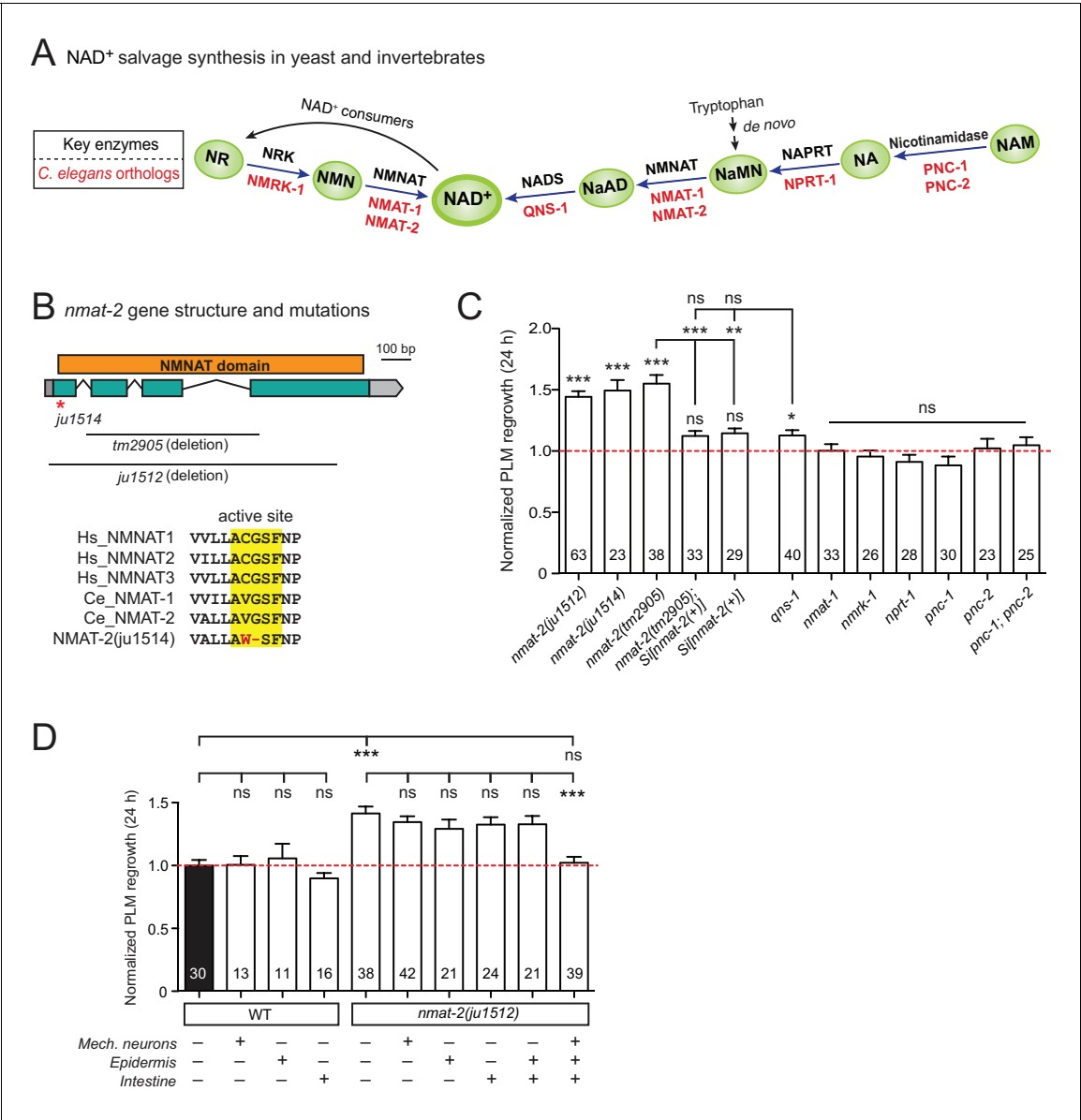

**Figure 2.** NMNAT/NMAT-2 inhibits PLM axon regrowth via its catalytic domain. (A) Overview of NAD+ salvage biosynthesis pathway. Top, key enzymes; Bottom, *C. elegans* orthologs (*Shaye and Greenwald, 2011*). (B) Top, *nmat-2* gene structure and mutant alleles. NMAT-2 contains an NMNAT domain. *nmat-2(ju1514)* point mutation and *nmat-2(ju1512)* deletion alleles were generated using CRISPR-Cas9 genome editing. Bottom, sequence alignment of the active site of NMNAT domain of *C. elegans* NMAT-2 (accession number: NP_492480.1; amino acids 4–14) with human NMNAT1–3 (NP_073624.2, NP_055854, NP_001307441) and *C. elegans* NMAT-1 (NP_510010.2). Sequences were analyzed using Clustal Omega. (C) Normalized regrowth 24 hr post-axotomy in mutants lacking genes encoding enzymes in the NAD+ biosynthesis pathway. Statistics, Student's *t*-test with same day controls. For the statistical test of transgene analysis, one-way ANOVA followed by Tukey's multiple comparison test. (D) PLM axon regrowth 24 hr post-axotomy in transgenic animals expressing *nmat-2(+)* driven by tissue-specific promoters for mechanosensory neurons (*Pmec-4*), epidermis (*Pcol-12*) or intestine (*Pmtl-2*) in a *nmat-2(ju1512)* background. One-way ANOVA followed by Tukey's multiple comparison test. Data are shown as mean ± SEM. n, number of animals shown within columns. ns, not significant; *p<0.05; **p<0.01; ***p<0.001.

DOI: https://doi.org/10.7554/eLife.39756.009

The following source data and figure supplement are available for figure 2:

**Source data 1.** Each data point in *Figure 2C,D*.
DOI: https://doi.org/10.7554/eLife.39756.011

**Figure supplement 1.** *nmat-1* show no defect in PLM axon regrowth even when its germline is defective.
DOI: https://doi.org/10.7554/eLife.39756.010

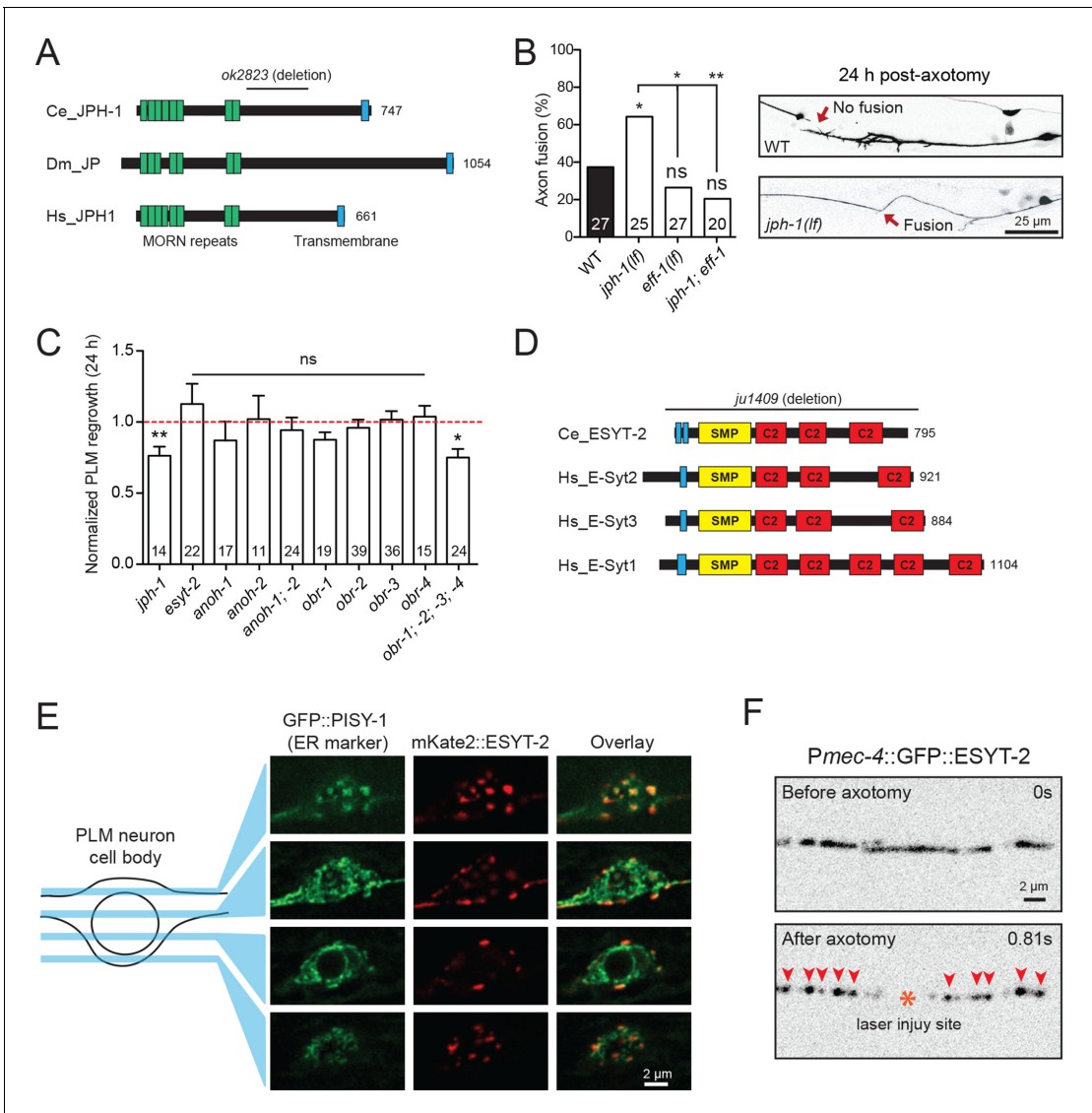

**Figure 3.** Select ER-PM membrane contact site proteins are required for axon regeneration and are sensitive to injury. (**A**) Junctophilin-1 protein structure. From top to bottom: *C. elegans* JPH-1 (NP_492193.2), its *Drosophila* ortholog JP (NP_523525.2), and human ortholog JPH1 (NP_065698.1). Junctophilins contain N-terminal MORN (Membrane Occupation and Recognition Nexus) repeats (green) and a C-terminal transmembrane domain (blue). *C. elegans* deletion allele is indicated above (*ok2823*). (**B**) Percentage of axons that exhibit fusion between the regrowing axon and distal fragment 24 hr post-axotomy. Upper image shows a regrowing axon that has not fused with the distal fragment in a wild-type animal. Lower image shows fusion between the regrowing axon and the distal fragment in a *jph-1(ok2823)* animal. Fisher's exact test. *$p<0.05$; **$p<0.01$. (**C**) Normalized regrowth 24 hr post-axotomy in mutants lacking selected genes encoding ER-PM MCS proteins. Data are shown as mean ± SEM. n, number of animals shown within columns. Student's *t*-test with same day controls. ns, not significant; *$p<0.05$; **$p<0.01$. (**D**) E-Syt protein structure. From top to bottom: *C. elegans* ESYT-2 and its human orthologs E-Syt2, E-Syt3, and E-Syt1 (NP_065779.1, NP_114119.2, NP_056107.1, respectively). Amino acid length is indicated to the right of each protein. E-Syt proteins contain N-terminal hydrophobic regions (blue), SMP (Synaptotagmin-like Mitochondrial and lipid-binding Protein) domains (yellow), and C-terminal C2 domains (red). *C. elegans* deletion allele is indicated above (*ju1409*). (**E**) Images of the PLM cell body and surrounding neurites. Left, GFP::PISY-1 ER marker; Middle, mKate2::ESYT-2 driven by the *mec-4* promoter; Right, Image overlays. Images show single slices taken at 1 μm intervals. (**F**) Representative inverted grayscale images of GFP::ESYT-2 in the axon of the PLM neuron before and immediately after axotomy (upper and lower panels, respectively). Site of laser axotomy indicated by asterisk; puncta indicated by arrowheads.
DOI: https://doi.org/10.7554/eLife.39756.012

The following source data and figure supplement are available for figure 3:

**Source data 1.** Each data point in *Figure 3C*.
DOI: https://doi.org/10.7554/eLife.39756.014

**Figure supplement 1.** *esyt-2* is widely expressed in the nervous system.
DOI: https://doi.org/10.7554/eLife.39756.013

condensed into small puncta almost immediately (<1 s) (*Figure 3F*; lower panel). As axon injury triggers a rapid rise in axonal calcium (*Ghosh-Roy et al., 2010*), we speculate that the injury-induced Ca$^{2+}$ transient triggers ESYT-2 relocalization to axonal ER-PM contact sites. This is consistent with the observation that vertebrate E-Syt1 can localize to ER-PM contact sites following an increase in cytosolic calcium (*Giordano et al., 2013*; *Idevall-Hagren et al., 2015*). We generated *esyt-2* null mutants by genome editing (*Figure 3D*). These mutant animals were indistinguishable from wild-type animals in growth rate, body morphology, and exhibited normal axon development and regrowth (*Figure 3C*). Thus, while ESYT-2 undergoes temporal changes in response to axon injury, it does not appear to be essential for axon regrowth.

The Anoctamin protein family function as tethers at ER-PM contact sites in yeast (*Manford et al., 2012*; *Wolf et al., 2012*). *C. elegans* has two orthologs, ANOH-1 and ANOH-2. ANOH-1 is expressed in mechanosensory neurons and acts together with the apoptotic factor CED-7 to promote phosphatidylserine exposure in the removal of necrotic cells (*Li et al., 2015*). *ced-7(0)* reduces PLM axon regrowth (*Neumann et al., 2015*). However, we found that loss of function in *anoh-1* or *anoh-2*, or the *anoh-1; anoh-2* double mutant, did not affect PLM axon regeneration (*Figure 3C*).

The eukaryotic OSBP and OSBP-related (ORP) family of MCS-localized lipid transfer proteins includes multiple members. ORP5/8 act as tethers at ER-PM MCSs where they mediate PI4P/Phosphatidylserine counter-transport, while OSBP and the other ORPs function at different MCSs (*Chung et al., 2015*). We tested the four *C. elegans* homologs individually as well as a quadruple mutant. Each *obr* single mutant displayed normal regeneration, and the quadruple mutant displayed a significant decrease in axon regrowth (*Figure 3C*). While the expression pattern and action site of these OBR proteins remain to be determined, our finding is consistent with the known redundancy within the OBR family (*Kobuna et al., 2010*).

Altogether, the above analysis echoes a recent study in yeast where elimination of multiple MCS components did not impair ER-PM sterol exchange (*Quon et al., 2018*), highlighting the challenge to tease apart the functional redundancy of MCS proteins in biological processes.

## Lipid metabolic enzymes likely have extensive functional redundancy in axon regrowth

Lipids are essential components of membranes and regulate many biological functions including energy storage and lipid signaling. In *C. elegans*, the majority of triglyceride is obtained from the diet, and lipogenesis accounts for less than 10% of stored body fat (*Srinivasan, 2015*). Lipolysis is required for cellular uptake or release of fatty acids and glycerol (*Zechner et al., 2012*). Classical 'neutral' lipolysis involves at least three different lipases: ATGL (adipose triglyceride lipase), HSL (hormone sensitive lipase), and MGL (monoglyceride lipase). ATGL requires a coactivator protein, CGI-58/ABHD5. *C. elegans* encodes a single ATGL (ATGL-1), three CGI-58/ABHD5 (ABHD-5.2, ABHD-5.3, and LID-1), a single HSL (HOSL-1), but lacks MGL by sequence homology (*Zechner et al., 2012*). We tested single mutants for all these genes and double or triple mutants for ABHD (α/β hydrolase domain) genes and observed no detectable effects in PLM axon regrowth (*Figure 4A*).

Triglycerides can also be hydrolyzed through autophagy-mediated degradation of lipid droplets by some lysosomal acid lipases, termed lipophagy or 'acid' lipolysis (*Singh et al., 2009*). *C. elegans* lysosomal lipases (LIPL-1, LIPL-3, and LIPL-4), autophagy proteins (LGG-1 and LGG-2), and transcription factors (HLH-30/TFEB and MXL-3/MXL) act in lipophagy (*Folick et al., 2015*; *O'Rourke and Ruvkun, 2013*). Two nuclear hormone receptors NHR-49/PPARα and NHR-80/HNF4α are reported to regulate LIPL-4 (*Folick et al., 2015*). We found that single mutants for all these genes showed normal PLM regrowth (*Figure 4B*), suggesting that lipolysis may not play an essential role in PLM axon regeneration.

The Kennedy pathway synthesizes the most abundant phospholipids in eukaryotic membranes, phosphatidylcholine (PC) and phosphatidylethanolamine (PE) (*Gibellini and Smith, 2010*), and involves conserved enzymes catalyzing a series of consecutive reactions (*Figure 4C*). Of all mutants affecting individual enzymes in the Kennedy pathway, we found that *cept-2* null mutants showed a significant reduction in axon regrowth (*Figure 4D*). In testing functional redundancy between *cept-1* and *cept-2*, we found double mutants to be embryonic or larval lethal (data not shown), preventing further analysis. Definitive conclusions will require tissue-specific and temporal manipulation of this pathway. Overall, our analysis suggests that the Kennedy pathway may affect axon regeneration.

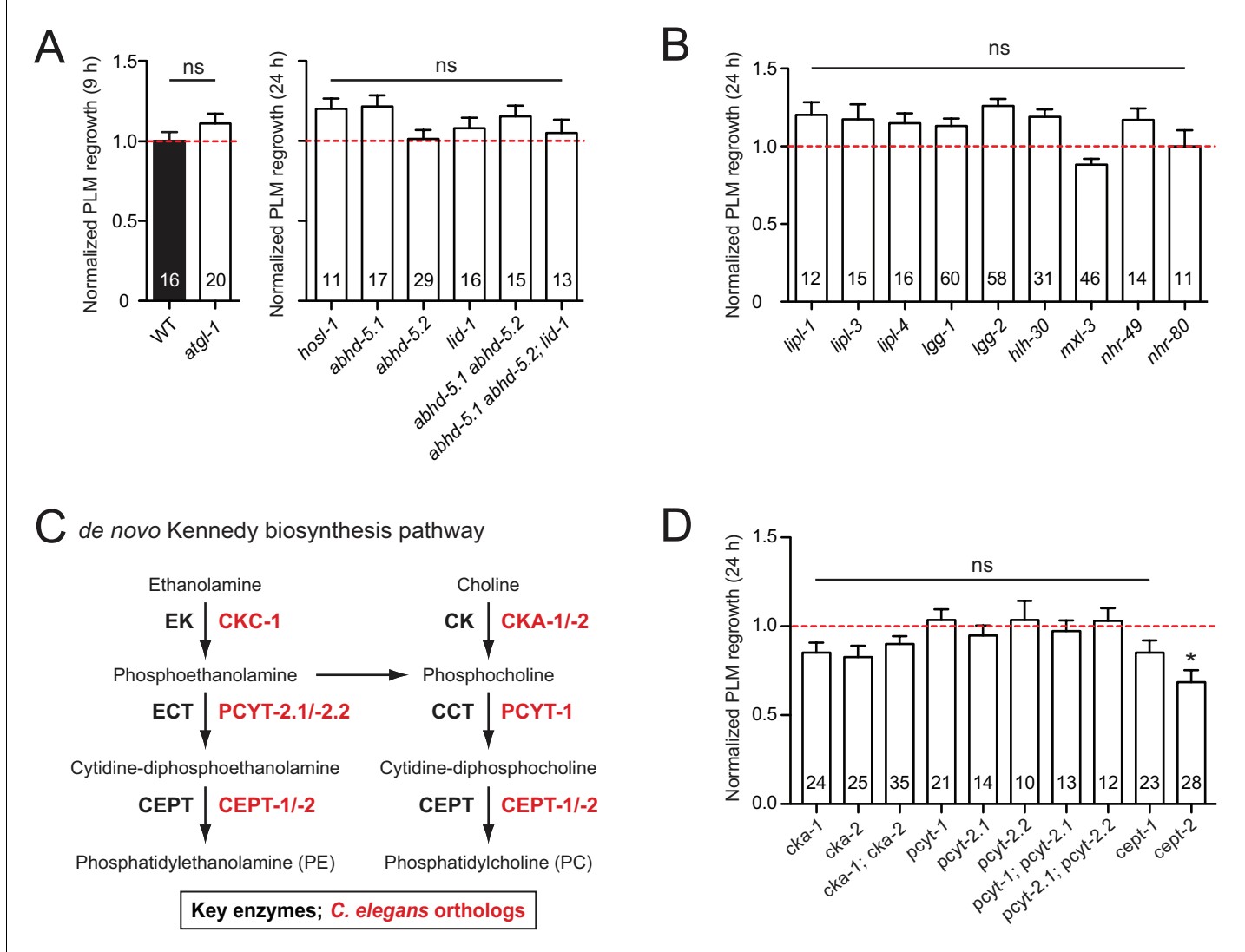

**Figure 4.** PLM axon regeneration involves membrane lipid biosynthesis pathway. (A) Normalized PLM axon regrowth 24 hr post-axotomy in mutants affecting neutral lipolysis. (B) Normalized PLM axon regrowth 24 hr post-axotomy in mutants affecting acid lipolysis. (C) Overview of *C.elegans* Kennedy pathway for de novo biosynthesis of PE and PC, the major phospholipids in the PM. (D) Normalized PLM axon regrowth 24 hr post-axotomy in mutants lacking select genes encoding enzymes in the Kennedy pathway. Data are shown as mean ±SEM. n, number of animals shown within columns. Student's *t*-test with same day controls. ns, not significant; *p<0.05.

DOI: https://doi.org/10.7554/eLife.39756.015

The following source data is available for figure 4:

**Source data 1.** Each data point in *Figure 4A,B,D*.
DOI: https://doi.org/10.7554/eLife.39756.016

## The conserved NS1A-BP ortholog IVNS-1 inhibits axon regrowth

Among other conserved proteins, we identified the BTB-Kelch family protein IVNS-1 (Influenza Virus NS1A binding protein/NS1A-BP) as an inhibitor of axon regrowth. BTB/POZ (Broad-Complex, Tramtrack, and Bric-a-Brac/Poxvirus and Zinc finger) domain and Kelch repeats function in a wide variety of biological processes including gene expression, protein ubiquitination, and cytoskeleton binding (*Dhanoa et al., 2013*). Human NS1A-BP was originally identified based on interaction with the influenza A virus via its Kelch domain (*Wolff et al., 1998*) (*Figure 5A*) and was later found to interact with actin filaments (*Perconti et al., 2007*) and RNA binding proteins, including heterogeneous

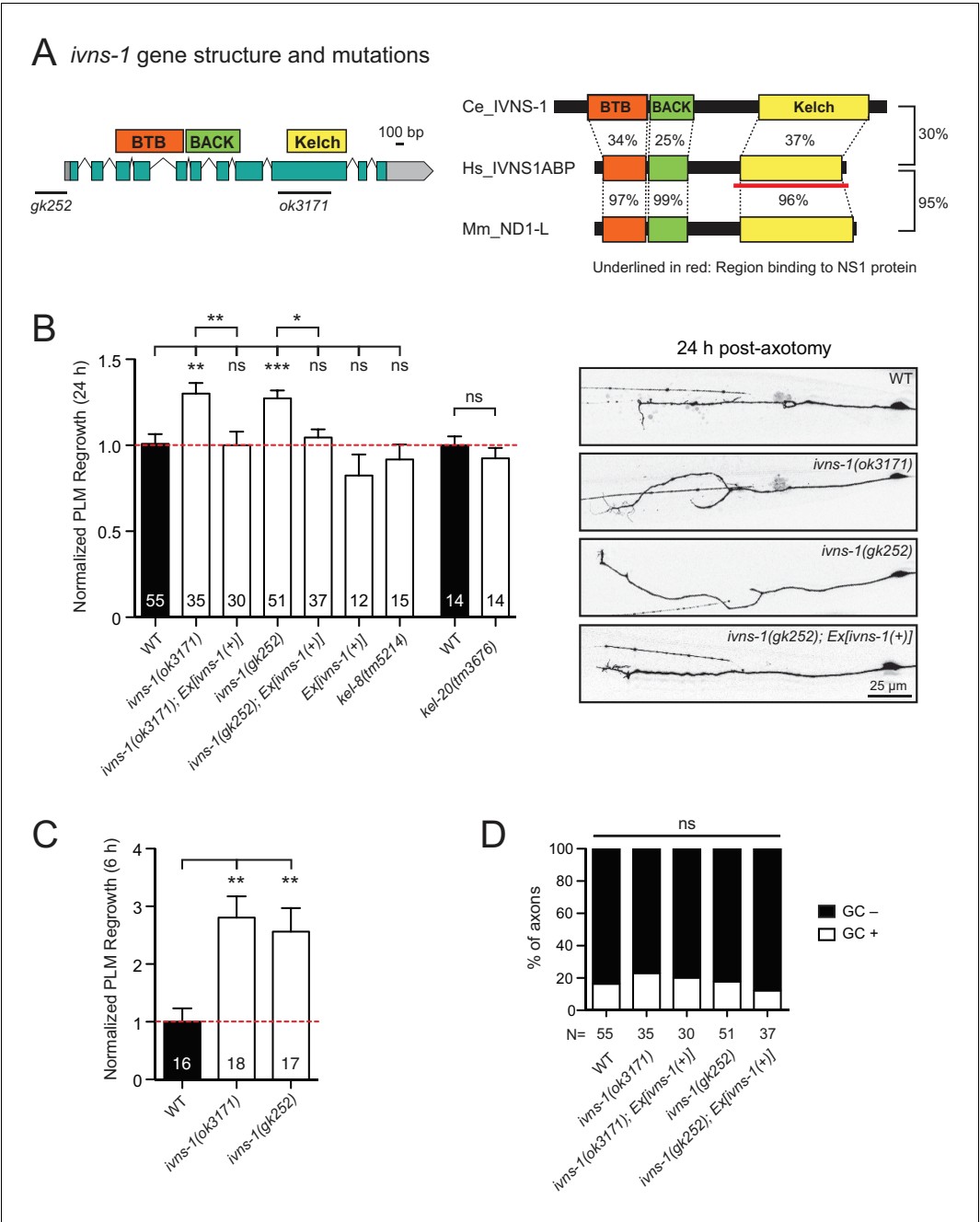

**Figure 5.** The Kelch-domain protein IVNS-1 inhibits axon regeneration. (**A**) *ivns-1* gene structure. Left: Loss-of-function alleles are indicated below (*gk252* and *ok3171*). Right: Alignment of the *C. elegans* IVNS-1 (NP_510109.1) with its human ortholog IVNS1ABP (NP_006460.1) and mouse ortholog ND1-L (NP_473443.2). Number indicates percentage identity of protein sequences. Sequences were analyzed using Clustal Omega. (**B**) Normalized PLM axon regrowth 24 hr post-axotomy in mutants of Kelch-domain proteins. Data are shown as mean ±SEM. n, number of animals shown within columns. Student's *t*-test with same day controls. ns, not significant; *$p<0.05$; **$p<0.01$; ***$p<0.001$. Right: representative inverted grayscale images of PLM 24 hr post-axotomy. Scale bar, 25 µm. (**C**) Normalized PLM axon regrowth 6 hr post-axotomy. Data are shown as mean ±SEM. One-way ANOVA followed by Tukey's multiple comparison test. n, number of animals shown within columns. **$p<0.01$. (**D**) Percentage of axons with growth cones (GCs) 6 hr post-axotomy. n, Number of animals shown below columns. Fisher's exact test. ns, not significant.

DOI: https://doi.org/10.7554/eLife.39756.017

The following source data is available for figure 5:

**Source data 1.** Each data point in *Figure 5B,C*.
DOI: https://doi.org/10.7554/eLife.39756.018

nuclear ribonucleoprotein and splicing factors (hnRNPs) and RNA helicase (*Tsai et al., 2013*). *C. elegans* IVNS-1 has the same overall domain organization as NS1A-BP (*Figure 5A*).

We analyzed two independent *ivns-1* mutants (*gk252* and *ok3171*) and observed increased axon regrowth, which was restored to control levels following transgenic expression of *ivns-1* driven by its own promoter (*Figure 5A,B*). *ivns-1* mutants showed increased regrowth as early as 6 hr post-injury (*Figure 5C*), while growth cone formation in *ivns-1* mutants was normal (*Figure 5D*). The effects of *ivns-1* on axon regrowth appeared to be unique, as mutants in two other BTB-Kelch proteins *kel-8* and *kel-20* displayed normal regrowth (*Figure 5B*). Whether the function of IVNS-1 involves actin cytoskeleton or RNA regulation remains to be determined.

## Overview of common themes

### Complex roles of basement membrane ECM and ADAMTSs

ECM plays diverse roles in axon regeneration (*Barros et al., 2011*). In *C. elegans*, neuronal processes are closely associated with basement membrane (BM) (*White et al., 1976*), which is a thin, specialized ECM adjacent to epithelial tissues (*Jayadev and Sherwood, 2017*). We previously reported that BM components SPON-1/F-spondin and PXN-2/Peroxidasin inhibit axon regrowth (*Chen et al., 2011*; *Gotenstein et al., 2010*). We further analyzed mutants of essential BM structural components and found that a loss-of-function mutant (*gm121*) of EPI-1/Laminin α and a gain-of-function mutant (*tk75*) of EMB-9/Type IV collagen (*Kubota et al., 2012*) both showed enhanced regrowth (*Figure 1—figure supplement 2A*), supporting defined roles of specific BM components in axon regrowth.

ADAMTS proteins are secreted metalloproteases and act as key ECM remodeling enzymes (*Tang, 2001*). In the mammalian nervous system, ADAMTS4 promotes axon regeneration and recovery after spinal cord injury, by digesting chondroitin sulfate proteoglycans (CSPGs), which are known to be prominent inhibitory components of the glial scar (*Tauchi et al., 2012*). In *C. elegans*, multiple chondroitin proteoglycans are expressed, but are not sulfated (*Olson et al., 2006*). The PLM axon is enveloped by the surrounding epidermis (*Emtage et al., 2004*) and is not in direct contact with the BM after embryogenesis. However, regrowing PLM axons may come in contact with the BM during regrowth. We tested null mutants in all five ADAMTS homologs and found that ADT-1 and ADT-3 promote and MIG-17 inhibits PLM axon regrowth (*Figure 1—figure supplement 2A*). These results suggest opposing roles for ADAMTS family members in PLM axon regrowth. ADT-1 and ADT-3 may normally degrade inhibitory BM such that their deficiency leads to elevated BM and impairs axonal regrowth. In contrast, MIG-17 may degrade permissive BM (for example, Type IV collagen) such that its deficiency leads to elevated stable Type IV collagen and enhances axonal regrowth. Together, these data indicate the complex roles of ECM components and ADAMTSs.

### Permissive role of Rab GTPase RAB-8 and inhibitory role of annexin proteins NEX-1 and NEX-2 in axon regeneration

We previously showed that genes implicated in endocytosis of synaptic vesicles (e.g., *unc-57/endophilin*) or membrane trafficking (*rsef-1/RASEF*) are required for axon regeneration (*Chen et al., 2011*). Here, we tested additional membrane-trafficking factors, especially the Rab small GTPases. Trafficking of secretory vesicles from the Golgi is partly regulated by Rab8 (*Stenmark, 2009*), and trafficking of recycling endosomes is regulated by Rab11 (*Ascaño et al., 2009*). Lack of Rab8 results in decreased neurite outgrowth in embryonic hippocampal neurons (*Huber et al., 1995*). *C. elegans* RAB-8 has been implicated in membrane trafficking in ciliated neurons (*Kaplan et al., 2010*). We found that RAB-8 was required for PLM axon regrowth, whereas RAB-11.2 showed no impact (*Figure 1—figure supplement 2B*), suggesting that post-Golgi vesicle trafficking, rather than endosome recycling, may be important for axon regrowth.

The Annexins are calcium-dependent phospholipid-binding proteins (*Monastyrskaya et al., 2007*) with a wide variety of roles in membrane biology (*Mirsaeidi et al., 2016*) and plasma membrane repair/resealing (*Boye and Nylandsted, 2016*). *C. elegans* has four Annexins (NEX-1/−2/−3/−4) (*Daigle and Creutz, 1999*); and NEX-1 was shown to promote GABAergic motor neuron regeneration (*Nix et al., 2014*). We found that NEX-1 and NEX-2 have an inhibitory role on PLM regrowth, whereas NEX-3 and NEX-4 have no impact (*Figure 1—figure supplement 2B*). These results suggest cell-type-dependent roles of Annexins in regrowth.

## Further evidence for permissive roles of the MT cytoskeleton in axon regeneration

Precise regulation of MT dynamics is a critical factor in axon regrowth (*Blanquie and Bradke, 2018*; *Tang and Chisholm, 2016*). Our previous studies identified EFA-6 as an intrinsic inhibitor of regrowth by acting as an axonal MT-destabilizing factor (*Chen et al., 2015*; *Chen et al., 2011*). We and others have also reported that MT post-translational modifications have differential roles in axon regeneration (*Cho and Cavalli, 2012*; *Ghosh-Roy et al., 2012*). MT stabilization is linked to acetylation of α-tubulins (*Janke and Montagnac, 2017*) and has been shown to improve regrowth; for example, pharmacological MT stabilization by Paclitaxel or Epothilone B promotes axon regrowth in multiple models (*Chen et al., 2011*; *Ruschel et al., 2015*; *Sengottuvel et al., 2011*). Here, we tested two α-tubulin acetyltransferases, MEC-17 and ATAT-2, which acetylate the α-tubulin MEC-12 that is enriched in mechanosensory neurons (*Akella et al., 2010*). We found that MEC-17, but not ATAT-2, was required for normal axon regrowth (*Figure 1—figure supplement 2C*). *mec-17; atat-2* double mutants showed reduced axon regrowth resembling the *mec-17* single mutant (*Figure 1—figure supplement 2C*), suggesting that MEC-17-dependent acetylated MTs are permissive for axon regrowth. In addition, the HDAC orthologs HDA-3 and HDA-6 inhibit axon regrowth (*Chen et al., 2011*) (*Table 2*). HDAC family proteins, which can deacetylate MTs and other targets, have been shown to be involved in mammalian axon regeneration (*Cho and Cavalli, 2012*). Overall, our results support a pro-regenerative role for acetylated MTs in axon regrowth.

An increasing notion is that isotypes of tubulins influence MT composition and stability (*Tang and Jin, 2018*). PLM axons contain predominantly unusual 15 MT filaments made of MEC-7/β-tubulin and MEC-12/α-tubulin, and also express multiple tubulin isotypes (*Kaletsky et al., 2018*; *Lockhead et al., 2016*) that likely contribute to 11 protofilaments. We found that loss of function in *mec-12* or *tba-9* resulted in reduced regrowth (*Figure 1—figure supplement 2D*). In contrast, loss of function in *tba-7* showed enhanced regrowth. A recent study that examined the neurite growth of mechanosensory neurons has proposed that TBA-7/ α -tubulin likely functions as a destabilizing factor for MTs (*Zheng et al., 2017*). Our observation is consistent with this proposal, and supports the general role of stabilized MTs in promoting axon regrowth.

## Roles for actin filament regulators in axon regeneration

Growth cone formation is an important initial stage of axon regeneration and involves extensive remodeling of actin filaments (*Gomez and Letourneau, 2014*). Actin-binding proteins can promote actin filament assembly or disassembly, for example, Gelsolin severs actin filaments to promote disassembly (*Klaavuniemi et al., 2008*; *Liu et al., 2010*), while Twinfilin binds to the ADP-actin monomers and prevents their assembly into filaments (*Moseley et al., 2006*; *Palmgren et al., 2002*). Of the three Gelsolin-related proteins in *C. elegans*, *gsnl-1* and *viln-1* null mutants showed normal regrowth while partial loss-of-function mutants of *fli-1* displayed reduced PLM axon regrowth (*Figure 1—figure supplement 2E*). In contrast, lack of the Twinfilin homolog TWF-2 increased axon regrowth. Although both Gelsolin and Twinfilin can promote actin filament disassembly, they may have differential roles in regenerating *C. elegans* axons.

## Novel ion channels and transporters involved in PLM axon regeneration

Neuronal activity plays a significant role in axon regeneration in vertebrates and invertebrates (*Chen et al., 2011*; *Ghosh-Roy et al., 2010*; *Lim et al., 2016*; *Tedeschi et al., 2016*). Our prior screen tested 54 genes encoding channels and transporters and overall was consistent with neuronal excitability promoting PLM regrowth. Here, we examined an additional 58 channel or transporter genes (*Figure 1—source data 1*). We found several new genes in which loss-of-function mutation results in enhanced regeneration, including the sodium-sensitive channel *tmc-1* (*Chatzigeorgiou et al., 2013*) and an acetylcholine receptor alpha subunit (CHRNA6) *lgc-12* (*Cohen et al., 2014*) (*Table 2*). Additionally, we found that MFSD-6, a member of the Major Facilitator Superfamily Domain (MFSD) family, promotes PLM regrowth (*Figure 1—figure supplement 2F*). MFSD family proteins have 10–12 transmembrane regions (*Yan, 2015*); some mediate nutrient transport across the blood-brain barrier (*Ceder et al., 2017*; *Perland et al., 2017*), but most are of unknown function. *C. elegans* MFSD-6 was previously identified as a regulator of motor circuit activity and *mfsd-6(0)* mutants are resistant to inhibitors of cholinesterase (*McCulloch et al., 2017*). *mfsd-*

6 is expressed in most neurons, including mechanosensory neurons (*Ogurusu et al., 2015*). Loss-of-function mutants of *mfsd-6* exhibited reduced axon regrowth, which was rescued by expressing wild type *mfsd-6* under a pan-neuronal promoter (*Figure 1—figure supplement 2F*). As mechanosensory neurons are not thought to be cholinergic, yet other mutants of cholinergic signaling (e.g. *cha-1*/ChAT, *unc-17*/VChAT) are defective in axon regrowth (*Chen et al., 2011*), these data suggest a possible neuronal, but cell non-autonomous, role for acetylcholine signaling to be permissive for PLM regrowth.

## Discussion

Functional screening for axon regeneration phenotypes is a powerful approach to identify novel regulators of axon regrowth after injury. *C. elegans* PLM axons exhibit robust response to injury, and therefore allow efficient screening of positive and negative regulators of regrowth. In this work we have nearly doubled the number of genes tested using genetic mutations and the PLM regeneration assay, taking the total number of genes screened to 1267. We expanded some gene classes previously analyzed in depth (e.g. kinases, ECM components, ion channels, and transporters) and have also specifically targeted several pathways not addressed in our earlier screen, such as $NAD^+$ biosynthesis, MCS components, lipid metabolism, and actin regulators. Interestingly, both MCS components and lipid metabolism tested display a high degree of genetic redundancy, such that single mutants only occasionally display regeneration defects, and compound mutant strains are required to assess functional requirements. Nevertheless, our findings suggest that ER-PM contact sites may be regulated by axon injury and that phospholipid synthesis may be critical for axon regeneration. Further work will be required to define whether these pathways play a role in lipid addition to the regrowing axon membrane or a more general signaling role.

Several axon regeneration screens have now been reported in *C. elegans* and the results may be compared to assess reproducibility and generalizability of the results. The present work and our prior screen (*Chen et al., 2011*) analyzed the effect of genetic mutations on PLM axon regeneration. In contrast, other studies have used RNAi or genetic mutants to analyze motor neuron regeneration (*Nix et al., 2014*). Differing results between the two screens (e.g. the opposite requirement for *nex-1* in PLM and motor neurons) may reflect cell-type-specific roles of the regulators in axon regeneration. A recent genome-wide in vitro axon regeneration screen in mouse cortical neurons revealed significant overlap with orthologous genes identified from *C. elegans* screen despite differences in neuron types, species, and experimental methods (*Chen et al., 2011*; *Nix et al., 2014*; *Sekine et al., 2018*), suggesting significant conservation of regenerative mechanisms.

Our screen approach is based on candidates and not random mutagenesis, and thus classical estimates of genetic saturation do not apply. However, it is notable that the frequency of positive and negative hits in the current screen does not differ from our previous screen. Our prior screen included many previously well-studied axon guidance and outgrowth pathways and thus might have been enriched for functionally important factors, but the present analysis suggests many genes not previously associated with the nervous system (e.g. *ptps-1*, *tep-1*, *brap-2*) also have functionally important roles in regrowth. One trend is that fewer mutants with dramatically reduced regrowth (<30% of wild type, such as *dlk-1*, *unc-75*, *sdn-1*) were identified, and thus the number of genes essential for initiation of regrowth may be limited. On the other hand, the present screen identified new mutants with drastically enhanced regrowth (>140% of wild type, such as *efa-6* and *pxn-2* from previous screen and *nmat-2* and *drag-1* from this screen). Interestingly, a recent genome-wide screen for enhanced regrowth in mouse cortical neurons reported a positive hit rate of 3% (*Sekine et al., 2018*), whereas we find 3.9% of genes displayed significantly elevated axon regrowth. The frequency of axon regrowth phenotypes may therefore be consistent across screening platforms.

## Materials and methods

**Key resources table**

*Continued on next page*

*Continued*

| Reagent type (species) or resource | Designation | Source or reference | Identifiers | Additional information |
|---|---|---|---|---|
| Reagent type (species) or resource | Designation | Source or reference | Identifiers | Additional information |
| Bacterial strain | *E. coli*: OP50 | Caenorhabditis Genetics Center | RRID: WB-STRAIN:OP50 | |
| Genetic reagent (*C. elegans*) | Strain wild type N2 | Caenorhabditis Genetics Center | RRID:WB-STRAIN:N2_ (ancestral) | |
| Genetic reagent (*C. elegans*) | *CZ10969: Pmec-7-GFP(muIs32) II* | | | Considered as 'WT' in many axotomy experiments |
| Genetic reagent (*C. elegans*) | *CZ10175: Pmec-4-GFP(zdIs5) I* | | | Considered as 'WT' in many axotomy experiments |
| Genetic reagent (*C. elegans*) | *CZ25411: nmat-2(ju1512) I/hT2 I, III; Pmec-7-GFP(muIs32) II* | | | |
| Genetic reagent (*C. elegans*) | *CZ25415: nmat-2(ju1514) I/hT2 I, III; Pmec-7-GFP(muIs32) II* | | | |
| Genetic reagent (*C. elegans*) | *CZ17633: nmat-2(tm2905) I/hT2 I, III; Pmec-7-GFP(muIs32) II* | | | |
| Genetic reagent (*C. elegans*) | *CZ24324: Pmec-7-GFP(muIs32) II; qns-1(ju1563) IV/mIs11 sd IV* | | | |
| Genetic reagent (*C. elegans*) | *CZ25642: Pmec-4-GFP(zdIs5) I; Pmec-7-GFP(muIs32) II* | | | |
| Genetic reagent (*C.elegans*) | *CZ25534: nmrk-1(ok2571) I; Pmec-7-GFP(muIs32) II* | | | |
| Genetic reagent (*C. elegans*) | *CZ24241: Pmec-4-GFP(zdIs5) I; nprt-1(tm6342) IV* | | | |
| Genetic reagent (*C. elegans*) | *CZ24242: Pmec-4-GFP(zdIs5) I; pnc-1(tm3502) IV* | | | |
| Genetic reagent (*C. elegans*) | *CZ24802: Pmec-4-GFP(zdIs5) I; pnc-2(tm6438) IV* | | | |
| Genetic reagent (*C. elegans*) | *CZ25466: Pmec-7-GFP(muIs32) II; juSi347[nmat-2 gDNA] IV* | | | |
| Genetic reagent (*C. elegans*) | *CZ25469: nmat-2(tm2905) I; Pmec-7-GFP(muIs32) II; juSi347[nmat-2 gDNA] IV* | | | |
| Genetic reagent (*C. elegans*) | *CZ26216: Pmec-7-GFP(muIs32) II; Ex[Pmec-4-nmat-2(juEx7834)]* | | | |
| Genetic reagent (*C. elegans*) | *CZ26217: Pmec-7-GFP(muIs32) II; Ex[Pmec-4-nmat-2(juEx7835)]* | | | |
| Genetic reagent (*C. elegans*) | *CZ26220: Pmec-7-GFP(muIs32) II; Ex[Pcol-12-nmat-2(juEx7838)]* | | | |
| Genetic reagent (*C. elegans*) | *CZ26221: Pmec-7-GFP(muIs32) II; Ex[Pcol-12-nmat-2(juEx7839)]* | | | |
| Genetic reagent (*C. elegans*) | *CZ26218: Pmec-7-GFP(muIs32) II; Ex[Pmtl-2-nmat-2(juEx7836)]* | | | |
| Genetic reagent (*C. elegans*) | *CZ26219: Pmec-7-GFP(muIs32) II; Ex[Pmtl-2-nmat-2(juEx7837)]* | | | |
| Genetic reagent (*C. elegans*) | *CZ26222: nmat-2(ju1512) I/hT2 I, III; Pmec-7-GFP(muIs32) II; Ex[Pmec-4-nmat-2(juEx7840)]* | | | |

*Continued on next page*

*Continued*

| Reagent type (species) or resource | Designation | Source or reference | Identifiers | Additional information |
|---|---|---|---|---|
| Genetic reagent (*C. elegans*) | *CZ26223: nmat-2(ju1512) I/hT2 I, III; Pmec-7-GFP(muIs32) II; Ex[Pmec-4-nmat-2(juEx7841)]* | | | |
| Genetic reagent (*C. elegans*) | *CZ26224: nmat-2(ju1512) I/hT2 I, III; Pmec-7-GFP(muIs32) II; Ex[Pcol-12-nmat-2(juEx7842)]* | | | |
| Genetic reagent (*C. elegans*) | *CZ26225: nmat-2(ju1512) I/hT2 I, III; Pmec-7-GFP(muIs32) II; Ex[Pcol-12-nmat-2(juEx7843)]* | | | |
| Genetic reagent (*C. elegans*) | *CZ26310: nmat-2(ju1512) I/hT2 I, III; Pmec-7-GFP(muIs32) II; Ex[Pmtl-2-nmat-2(juEx7836)]* | | | |
| Genetic reagent (*C. elegans*) | *CZ26311: nmat-2(ju1512) I/hT2 I, III; Pmec-7-GFP(muIs32) II; Ex[Pmtl-2-nmat-2(juEx7837)]* | | | |
| Genetic reagent (*C. elegans*) | *CZ26332: nmat-2(ju1512) I/hT2 I, III; Pmec-7-GFP(muIs32) II; Ex[Pmtl-2, col-12::nmat-2(juEx7853)]* | | | |
| Genetic reagent (*C. elegans*) | *CZ26333: nmat-2(ju1512) I/hT2 I, III; Pmec-7-GFP(muIs32) II; Ex[Pmtl-2, col-12::nmat-2(juEx7854)]* | | | |
| Genetic reagent (*C. elegans*) | *CZ26285: nmat-2(ju1512) I/hT2 I, III; Pmec-7-GFP(muIs32) II; Ex[Pmtl-2, col-12, mec-4::nmat-2(juEx7850)]* | | | |
| Genetic reagent (*C. elegans*) | *CZ26286: nmat-2(ju1512) I / hT2 I, III; Pmec-7-GFP(muIs32) II; Ex[Pmtl-2, col-12, mec-4::nmat-2(juEx7851)]* | | | |
| Genetic reagent (*C. elegans*) | *CZ26391: jph-1(ok2823) I; Pmec-7-GFP(muIs32) II* | | | |
| Genetic reagent (*C. elegans*) | *CZ22032: Pmec-4-GFP(zdIs5) I; anoh-1(tm4762) III* | | | |
| Genetic reagent (*C. elegans*) | *CZ22033: Pmec-4-GFP(zdIs5) I; anoh-2(tm4796) IV* | | | |
| Genetic reagent (*C. elegans*) | *CZ26325: Pmec-4-GFP(zdIs5) I; anoh-1(tm4762) III; anoh-2(tm4796) IV* | | | |
| Genetic reagent (*C. elegans*) | *CZ26069: Pmec-4-GFP(zdIs5) I; obr-1(xh16) III* | | | |
| Genetic reagent (*C. elegans*) | *CZ24555: Pmec-4-GFP(zdIs5) I; obr-2 (xh17) V* | | | |
| Genetic reagent (*C. elegans*) | *CZ24556: Pmec-4-GFP(zdIs5) I; obr-3(tm1087) X* | | | |
| Genetic reagent (*C. elegans*) | *CZ24557: obr-4(tm1567) I; Pmec-4-GFP(zdIs5) I* | | | |
| Genetic reagent (*C. elegans*) | *CZ25696: obr-4(tm1567) I; Pmec-4-GFP(zdIs5) I; obr-1(xh16) III; obr-2(xh17) V; obr-3(tm1087) X* | | | |
| Genetic reagent (*C. elegans*) | *CZ26375: Pmec-4-GFP(zdIs5) I; esyt-2(ju1409) III* | | | |
| Genetic reagent (*C. elegans*) | *CZ26570: juIs540 [Pmec-4-mKate2-ESYT-2]; juEx7807[Pmec-4-GFP-PISY-1]* | | | |
| Genetic reagent (*C. elegans*) | *CZ24897: juEx7604 [Pmec-4-GFP-ESYT-2]* | | | |
| Genetic reagent (*C. elegans*) | *CZ22087: Pmec-7-GFP(muIs32) II; atgl-1(tm3116) III / hT2 I, III* | | | |

*Continued on next page*

*Continued*

| Reagent type (species) or resource | Designation | Source or reference | Identifiers | Additional information |
|---|---|---|---|---|
| Genetic reagent (C. elegans) | CZ22536: Pmec-7-GFP(muIs32) II; hosl-1(gk278589) X | | | |
| Genetic reagent (C. elegans) | CZ22006: Pmec-7-GFP(muIs32) II; abhd-5.1(ok3722) V | | | |
| Genetic reagent (C. elegans) | CZ21968: Pmec-7-GFP(muIs32) II; abhd-5.2(ok3245) V | | | |
| Genetic reagent (C. elegans) | CZ22007: lid-1(gk575511) I; Pmec-7-GFP(muIs32) II | | | |
| Genetic reagent (C. elegans) | CZ22163: Pmec-7-GFP(muIs32) II; abhd-5.2(ok3245) V abhd-5.1(ju1282) V | | | |
| Genetic reagent (C. elegans) | CZ22166: lid-1(gk575511) I; Pmec-7-GFP(muIs32) II; abhd-5.2(ok3245) abhd-5.1(ju1282) V | | | |
| Genetic reagent (C. elegans) | CZ22686: Pmec-7-GFP(muIs32) II; lipl-1(tm1954) V | | | |
| Genetic reagent (C. elegans) | CZ22688: Pmec-7-GFP(muIs32) II; lipl-3(tm4498) V | | | |
| Genetic reagent (C. elegans) | CZ22535: Pmec-7-GFP(muIs32) II; lipl-4(tm4417) V | | | |
| Genetic reagent (C. elegans) | CZ24364: Pmec-4-GFP(zdIs5) I; lgg-1(tm3489) II/ mln1 II | | | |
| Genetic reagent (C. elegans) | CZ23325: Pmec-7-GFP(muIs32) II; lgg-2(tm6474) IV | | | |
| Genetic reagent (C. elegans) | CZ23322: Pmec-7-GFP(muIs32) II; hlh-30(tm1978) IV | | | |
| Genetic reagent (C. elegans) | CZ14408: Pmec-7-GFP(muIs32) II; mxl-3(ok1947) X | | | |
| Genetic reagent (C. elegans) | CZ22541: nhr-49(nr2041) I; Pmec-7-GFP(muIs32) II | | | |
| Genetic reagent (C. elegans) | CZ22510: Pmec-7-GFP(muIs32) II; nhr-80(tm1011) III | | | |
| Genetic reagent (C. elegans) | CZ25587: Pmec-4-GFP(zdIs5) I; cka-1(tm1241) IV | | | |
| Genetic reagent (C. elegans) | CZ25549: Pmec-4-GFP(zdIs5) I; cka-2(tm841) X | | | |
| Genetic reagent (C. elegans) | CZ25403: Pmec-4-GFP(zdIs5) I; cka-1(tm1241) IV; cka-2(tm841) X. | | | |
| Genetic reagent (C. elegans) | CZ25370: Pmec-4-GFP(zdIs5) I; pcyt-1(et9) X | | | |
| Genetic reagent (C. elegans) | CZ25790: Pmec-4-GFP(zdIs5) I; pcyt-2.1(gk440213) I | | | |
| Genetic reagent (C. elegans) | CZ25368: Pmec-4-GFP(zdIs5) I; pcyt-2.2(ok2179) X | | | |
| Genetic reagent (C. elegans) | CZ25992: Pmec-4-GFP(zdIs5) I; pcyt-2.1(gk440213) I; pcyt-2.2(ok2179) X | | | |
| Genetic reagent (C. elegans) | CZ26521: Pmec-4-GFP(zdIs5) I; pcyt-2.1(gk440213) I; pcyt-1(et9) X | | | |
| Genetic reagent (C. elegans) | CZ25369: Pmec-4-GFP(zdIs5) I; cept-1(et10) X | | | |
| Genetic reagent (C. elegans) | CZ26423: Pmec-4-GFP(zdIs5) I; cept-2(ok3135) V | | | |
| Genetic reagent (C. elegans) | CZ19835: Pmec-4-GFP(zdIs5) I; kel-8(tm5214) V | | | |

*Continued*

| Reagent type (species) or resource | Designation | Source or reference | Identifiers | Additional information |
|---|---|---|---|---|
| Genetic reagent (C. elegans) | CZ23911: kel-20(tm3676) I; mec-7-GFP(muIs32) II | | | |
| Genetic reagent (C. elegans) | CZ18224: Pmec-4-GFP(zdIs5) I; ivns-1(ok3171) X | | | |
| Genetic reagent (C. elegans) | CZ18225: Pmec-4-GFP(zdIs5) I; ivns-1(gk252) X | | | |
| Genetic reagent (C. elegans) | CZ25508: Pmec-4-GFP(zdIs5) I; ivns-1(ok3171) X; Ex[ivns-1_gDNA(juEx7673)] | | | |
| Genetic reagent (C. elegans) | CZ25509: Pmec-4-GFP(zdIs5) I; ivns-1(ok3171) X; Ex[ivns-1_gDNA(juEx7674)] | | | |
| Genetic reagent (C. elegans) | CZ25510: Pmec-4-GFP(zdIs5) I; ivns-1(gk252) X; Ex[ivns-1_gDNA(juEx7673)] | | | |
| Genetic reagent (C. elegans) | CZ25511: Pmec-4-GFP(zdIs5) I; ivns-1(gk252) X; Ex[ivns-1_gDNA(juEx7674)] | | | |
| Genetic reagent (C. elegans) | CZ24755: juEx7584[Pesyt-2-GFP] | | | |
| Genetic reagent (C. elegans) | CZ21465: Pmec-4-GFP(zdIs5) I; epi-1(gm121) IV | | | |
| Genetic reagent (C. elegans) | CZ21463: Pmec-7-GFP(muIs32) II; emb-9(tk75) III | | | |
| Genetic reagent (C. elegans) | CZ21198: Pmec-7-GFP(muIs32) II; adt-1(cn30) X | | | |
| Genetic reagent (C. elegans) | CZ20937: Pmec-4-GFP(zdIs5) I; adt-2(wk156) X | | | |
| Genetic reagent (C. elegans) | CZ21004: Pmec-4-GFP(zdIs5) I; adt-3 (T19D2.1) (ok923) X | | | |
| Genetic reagent (C. elegans) | CZ26611: Pmec-7-GFP(muIs32) II; gon-1(e1254) IV / + | | | |
| Genetic reagent (C. elegans) | CZ23908: rab-8(tm2526) I; Pmec-7-GFP(muIs32) II | | | |
| Genetic reagent (C. elegans) | CZ23909: rab-11.2(tm2081) I; Pmec-7-GFP(muIs32) II | | | |
| Genetic reagent (C. elegans) | CZ20682: Pmec-7-GFP(muIs32) II; nex-1(gk148) III | | | |
| Genetic reagent (C. elegans) | CZ20683: Pmec-7-GFP(muIs32) II; nex-2(ok764) III | | | |
| Genetic reagent (C. elegans) | CZ20684: Pmec-7-GFP(muIs32) II; nex-3(gk385) III | | | |
| Genetic reagent (C. elegans) | CZ20685: Pmec-7-GFP(muIs32) II; nex-4(gk102) V | | | |
| Genetic reagent (C. elegans) | CZ14006: Pmec-7-GFP(muIs32) II; mec-17(ok2109) IV | | | |
| Genetic reagent (C. elegans) | CZ14008: Pmec-7-GFP(muIs32) II; atat-2(ok2415) X | | | |
| Genetic reagent (C. elegans) | CZ14848: mec-17(ok2109) IV; atat-2(ok2415) X; Pmec-7-GFP(muIs32) II | | | |
| Genetic reagent (C. elegans) | CZ17720: Pmec-7-GFP(muIs32) II; mec-12(tm5083) III | | | |
| Genetic reagent (C. elegans) | CZ9247: tba-1(ok1135) I; Pmec-7-GFP(muIs32) II | | | |

*Continued on next page*

*Continued*

| Reagent type (species) or resource | Designation | Source or reference | Identifiers | Additional information |
|---|---|---|---|---|
| Genetic reagent (*C. elegans*) | CZ26688: Pmec-4-GFP(zdIs5) I; tba-7(gk787939) III | | | |
| Genetic reagent (*C. elegans*) | CZ26833: Pmec-4-GFP(zdIs5) I; tba-7(u1015) III | | | |
| Genetic reagent (*C. elegans*) | CZ26635: Pmec-4-GFP(zdIs5) I; mec-7(ok2152) X | | | |
| Genetic reagent (*C. elegans*) | CZ10615: Pmec-4-GFP(zdIs5) I; tbb-2(gk129) III | | | |
| Genetic reagent (*C. elegans*) | CZ11083: Pmec-4-GFP(zdIs5) I; tbb-4(ok1461) X | | | |
| Genetic reagent (*C. elegans*) | CZ10810: Pmec-4-GFP(zdIs5) I; tbb-6(tm2004) V | | | |
| Genetic reagent (*C. elegans*) | CZ21461: Pmec-4-GFP(zdIs5) I; fli-1(ky535) III | | | |
| Genetic reagent (*C. elegans*) | CZ21199: Pmec-7-GFP(muIs32) II; gsnl-1(ok2979) V | | | |
| Genetic reagent (*C. elegans*) | CZ10888: viln-1(ok2413) I; Pmec-7-GFP(muIs32) II | | | |
| Genetic reagent (*C. elegans*) | CZ13606: Pmec-4-GFP(zdIs5) I; twf-2(ok3564) X | | | |
| Genetic reagent (*C. elegans*) | CZ20063: Pmec-4-GFP(zdIs5) I; mfsd-6(ju833) III | | | |
| Genetic reagent (*C. elegans*) | CZ19827: Pmec-4-GFP(zdIs5) I; mfsd-6(tm3356) III | | | |
| Genetic reagent (*C. elegans*) | CZ24417: Pmec-4-GFP(zdIs5) I; mfsd-6(tm3356) III; Prgef-1-mfsd-6(juEx6079) | | | |
| Genetic reagent (*C. elegans*) | CZ21030: Pmec-4-GFP(zdIs5) I; C05D9.7(ok2931) X | | | |
| Genetic reagent (*C. elegans*) | CZ25317: Pmec-4-GFP(zdIs5) I; dpy-10(e128) II | | | |
| Genetic reagent (*C. elegans*) | CZ23667: Pmec-4-GFP(zdIs5) I; F35G2.1(ok1669) IV | | | |
| Genetic reagent (*C. elegans*) | CZ23772: Pmec-4-GFP(zdIs5) I; gly-2(tm839) I | | | |
| Genetic reagent (*C. elegans*) | CZ17890: Pmec-4-GFP(zdIs5) I; osm-11(rt142) X | | | |
| Genetic reagent (*C. elegans*) | CZ17021: Pmec-4-GFP(zdIs5) I; zig-1(ok784) II | | | |
| Genetic reagent (*C. elegans*) | CZ17023: Pmec-4-GFP(zdIs5) I; zig-3(gk33) X | | | |
| Genetic reagent (*C. elegans*) | CZ17024: Pmec-4-GFP(zdIs5) I; zig-3(ok1476) X | | | |
| Genetic reagent (*C. elegans*) | CZ22031: abts-1(ok1566) I; Pmec-7-GFP(muIs32) II | | | |
| Genetic reagent (*C. elegans*) | CZ21461: Pmec-4-GFP(zdIs5) I; fli-1(ky535) III | | | |
| Genetic reagent (*C. elegans*) | CZ17435: Pmec-7-GFP(muIs32) II; mec-12(e1605) III | | | |
| Genetic reagent (*C. elegans*) | CZ17637: Pmec-4-GFP(zdIs5) I; tba-9(ok1858) X | | | |
| Genetic reagent (*C. elegans*) | CZ20033: vab-10(e698) I; Pmec-7-GFP(muIs32) II | | | |

*Continued on next page*

*Continued*

| Reagent type (species) or resource | Designation | Source or reference | Identifiers | Additional information |
|---|---|---|---|---|
| Genetic reagent (*C. elegans*) | CZ17099: Pmec-4-GFP(zdIs5) I; plk-1(or683ts) III | | | |
| Genetic reagent (*C. elegans*) | CZ17285: Pmec-4-GFP(zdIs5) I; svh-2(tm737) X | | | |
| Genetic reagent (*C. elegans*) | CZ19343: Pmec-4-GFP(zdIs5) I; cpr-1(ok1344) V | | | |
| Genetic reagent (*C. elegans*) | CZ19200:dhhc-11(gk1105) I; Pmec-7-GFP(muIs32) II | | | |
| Genetic reagent (*C. elegans*) | CZ22823: Pmec-4-GFP(zdIs5) I; eat-3(tm1107) II | | | |
| Genetic reagent (*C. elegans*) | CZ16134: Pmec-4-GFP(zdIs5) I; npr-20(ok2575) II | | | |
| Genetic reagent (*C. elegans*) | CZ23845: Pmec-4-GFP(zdIs5) I; ptps-1(tm1984) I | | | |
| Genetic reagent (*C. elegans*) | CZ17995: supr-1(ju1118) I; Pmec-7-GFP(muIs32) II | | | |
| Genetic reagent (*C. elegans*) | CZ12031: Pmec-4-GFP(zdIs5) I; rsef-1(ok1356) X | | | |
| Genetic reagent (*C. elegans*) | CZ17629: Pmec-4-GFP(zdIs5) I; osm-7(tm2256) III | | | |
| Genetic reagent (*C. elegans*) | CZ17098: Pmec-4-GFP(zdIs5) I; par-2(or373ts) III | | | |
| Genetic reagent (*C. elegans*) | CZ18676: Pmec-4-GFP(zdIs5) I; rgl-1(ok1921) X | | | |
| Genetic reagent (*C. elegans*) | CZ20056: Pmec-7-GFP(muIs32) II; wdfy-3(ok912) IV | | | |
| Genetic reagent (*C. elegans*) | CZ19721: Pmec-4-GFP(zdIs5) I wdr-23(tm1817) I | | | |
| Genetic reagent (*C. elegans*) | CZ22063: Pmec-4-GFP(zdIs5) I; brap-2(tm5132) II | | | |
| Genetic reagent (*C. elegans*) | CZ21217: Pmec-4-GFP(zdIs5) I; brap-2(ok1492) II | | | |
| Genetic reagent (*C. elegans*) | CZ19337: Pmec-4-GFP(zdIs5) I; cdc-48.1(tm544) II | | | |
| Genetic reagent (*C. elegans*) | CZ19725: Pmec-4-GFP(zdIs5) I; ced-9(n1950sd) III | | | |
| Genetic reagent (*C. elegans*) | CZ21651: Pmec-4-GFP(zdIs5) I; dnj-23(tm7102) II | | | |
| Genetic reagent (*C. elegans*) | CZ21356: Pmec-4-GFP(zdIs5) I; fbxc-50(tm5154) II | | | |
| Genetic reagent (*C. elegans*) | CZ16950: Pmec-7-GFP(muIs32) II; math-33(ok2974) V | | | |
| Genetic reagent (*C. elegans*) | CZ16951: Pmec-7-GFP(muIs32) II; skr-5(ok3068) V | | | |
| Genetic reagent (*C. elegans*) | CZ21010: tep-1(tm3720) I; Pmec-7-GFP(muIs32) II | | | |
| Genetic reagent (*C. elegans*) | CZ22796: mec-8(e398) I; Pmec-7-GFP(muIs32) II | | | |
| Genetic reagent (*C. elegans*) | CZ14510: Pmec-4-GFP(zdIs5) I; rict-1(mg360) II | | | |
| Genetic reagent (*C. elegans*) | CZ22570: rtcb-1(gk451) I / [bli-4(e937) let-?(q782) qIs48](hT2) I, III; Pmec-7-GFP(muIs32) II | | | |

*Continued on next page*

Continued

| Reagent type (species) or resource | Designation | Source or reference | Identifiers | Additional information |
|---|---|---|---|---|
| Genetic reagent (C. elegans) | CZ21655: Pmec-4-GFP(zdIs5) I; skn-1(ok2315) IV/nT1(qIs51) IV; V | | | |
| Genetic reagent (C. elegans) | CZ23377: Pmec-4-GFP(zdIs5) I; smg-3(r930) IV | | | |
| Genetic reagent (C. elegans) | CZ13997: Pmec-7-GFP(muIs32) II; syd-9(ju49) X | | | |
| Genetic reagent (C. elegans) | CZ21723: Pmec-4-GFP(zdIs5) I; tdp-1(ok803) II | | | |
| Genetic reagent (C. elegans) | CZ23133: Pmec-4-GFP(zdIs5) I; wdr-5.1(ok1417) III | | | |
| Genetic reagent (C. elegans) | CZ21194: Pmec-7-GFP(muIs32) II; mig-17(k174) V | | | |
| Genetic reagent (C. elegans) | CZ22792: Pmec-7-GFP(muIs32) II; ZC116.3(ok1618) V | | | |
| Genetic reagent (C. elegans) | CZ19193: Pmec-7-GFP(muIs32) II; lgc-12(ok3546) III | | | |
| Genetic reagent (C. elegans) | CZ18217: Pmec-4-GFP(zdIs5) I; tmc-1(ok1859) X | | | |
| Genetic reagent (C. elegans) | CZ17639: Pmec-4-GFP(zdIs5) I; nud-1(ok552) III | | | |
| Genetic reagent (C. elegans) | CZ17841: Pmec-4-GFP(zdIs5) I; mgl-1(tm1811) X | | | |
| Genetic reagent (C. elegans) | CZ17843: Pmec-4-GFP(zdIs5) I; mgl-3(tm1766) IV | | | |
| Genetic reagent (C. elegans) | CZ17848: Pmec-4-GFP(zdIs5) I; npr-25(ok2008) V | | | |
| Genetic reagent (C. elegans) | CZ22890: Pmec-4-GFP(zdIs5) I; ucr-2.3(ok3073) III | | | |
| Genetic reagent (C. elegans) | CZ15607: drag-1(tm3773) I; Pmec-7-GFP(muIs32) II | | | |
| Genetic reagent (C. elegans) | CZ17393: Pmec-4-GFP(zdIs5) I; snb-6(tm5195) II | | | |
| Genetic reagent (C. elegans) | CZ17018: Pmec-4-GFP(zdIs5) I; drag-1(tm3773) I | | | |
| Genetic reagent (C. elegans) | CZ18617: Pmec-4-GFP(zdIs5) I; ect-2(ku427) II | | | |
| Genetic reagent (C. elegans) | CZ18818: Pmec-4-GFP(zdIs5) I; lin-2(e1309) X | | | |
| Genetic reagent (C. elegans) | CZ18817: Pmec-4-GFP(zdIs5) I; magi-1(zh66) IV | | | |
| Genetic reagent (C. elegans) | CZ20673: Pmec-4-GFP(zdIs5) I; prmt-5(gk357) III | | | |
| Genetic reagent (C. elegans) | CZ18816: Pmec-4-GFP(zdIs5) I; rap-1(tm861) IV | | | |
| Genetic reagent (C. elegans) | CZ18460: Pmec-4-GFP(zdIs5) I; smz-1(ok3576) IV | | | |
| Genetic reagent (C. elegans) | CZ22544: Pmec-4-GFP(zdIs5) I; trxr-1(sv47) IV | | | |
| Genetic reagent (C. elegans) | CZ24963: Pmec-4-GFP(zdIs5) I; natb-1(ju1405) V/nT1 IV; V | | | |
| Genetic reagent (C. elegans) | CZ16946: Pmec-7-GFP(muIs32) II; rnf-5(tm794) III | | | |

*Continued*

| Reagent type (species) or resource | Designation | Source or reference | Identifiers | Additional information |
|---|---|---|---|---|
| Genetic reagent (*C. elegans*) | *CZ23068: ulp-5/tofu-3(tm3063) I; Pmec-7-GFP(muIs32) II* | | | |
| Genetic reagent (*C. elegans*) | *CZ23091: Pmec-7-GFP(muIs32) II; csr-1(fj54) IV/nT1 IV; V* | | | |
| Genetic reagent (*C. elegans*) | *CZ17638: Pmec-4-GFP(zdIs5) I; elpc-3(ok2452) V* | | | |
| Genetic reagent (*C. elegans*) | *CZ12938: Pmec-4-GFP(zdIs5) I; hda-6(tm3436) IV* | | | |
| Recombinant DNA reagent | Plasmid: pCZGY3260: *nmat-2* genomic DNA | This work | N/A | *nmat-2* genomic DNA (~1500 bp upstream; ~670 bp downstream); modified pCFJ201 plasmid for modified MosSCI on ChIV |
| Recombinant DNA reagent | Plasmid: pCZ993: *Pmec-4-nmat-2 gDNA-let-858 3'UTR* | This work | N/A | *nmat-2* expression driven by *mec-4* promoter in the mechanosensory neurons |
| Recombinant DNA reagent | Plasmid: pCZ994: *Pmtl-2-nmat-2 gDNA-let-858 3'UTR* | This work | N/A | *nmat-2* expression driven by *mtl-2* promoter in the intestine |
| Recombinant DNA reagent | Plasmid: pCZ995: *Pcol-12-nmat-2 gDNA-let-858 3'UTR* | This work | N/A | *nmat-2* expression driven by *col-12* promoter in the epiderdims |
| Recombinant DNA reagent | Plasmid: pCZGY3329: *Pmec-4-GFP-ESYT-2* | This work | N/A | |
| Recombinant DNA reagent | Plasmid: pCZGY3344: *Pmec-4-mKate2-ESYT-2* | This work | N/A | |
| Recombinant DNA reagent | Plasmid: pCZGY3342: *Pmec-4-mKate2-PISY-1* | This work | N/A | |
| Recombinant DNA reagent | Plasmid: pCZGY3302: *ivns-1* genomic DNA | This work | N/A | *ivns-1* genomic DNA (2 kb upstream; 800 bp downstream) |
| Recombinant DNA reagent | Plasmid: pCZGY3347: *Prgef-1-mfsd-6* | This work | N/A | |
| Recombinant DNA reagent | Plasmid: pCZGY3346: *Pesyt-2-GFP* | This work | N/A | GFP expression driven by *esyt-2* promoter |
| Sequence-based reagent | crRNA: nmat-2:/AltR1/rCrG rArGrU rCrGrC rUrCrU rUrCrU rUrGrC rCrGrU rGrUrU rUrUrA rGrArG rCrUrA rUrGrC rU/AltR2/ | IDT | N/A | crRNA to make *nmat-2(ju1512)* and *nmat-2(ju1514)* |
| Sequence-based reagent | crRNA: nmat-2:/AltR1/rCrG rUrGrU rUrGrA rArCrU rArArC rUrCrC rArCrU rGrUrU rUrUrA rGrArG rCrUrA rUrGrC rU/AltR2/ | IDT | N/A | crRNA to make *nmat-2(ju1512)* |

*Continued on next page*

*Continued*

| Reagent type (species) or resource | Designation | Source or reference | Identifiers | Additional information |
|---|---|---|---|---|
| Sequence-based reagent | crRNA: nmat-1:/AltR1/rArA rCrUrU rUrUrU rCrGrG rUrCrC rCrCrA rUrArG rGrUrU rUrUrA rGrArG rCrUrA rUrGrC rU/AltR2/ | IDT | N/A | crRNA to make nmat-1 (ju1565) |
| Sequence-based reagent | crRNA: nmat-1:/AltR1/rArU rGrUrA rCrUrU rGrArU rUrArC rGrGrA rArUrC rGrUrU rUrUrA rGrArG rCrUrA rUrGrC rU/AltR2/ | IDT | N/A | crRNA to make nmat-1(ju1565) |
| Sequence-based reagent | crRNA: qns-1:/AltR1/rGrG rUrGrU rUrArU rUrCrA rCrGrU rGrUrU rArCrA rGrUrU rUrUrA rGrArG rCrUrA rUrGrC rU/AltR2/ | IDT | N/A | crRNA to make qns-1(ju1563) |
| Sequence-based reagent | crRNA: qns-1:/AltR1/rGrA rUrArA rCrUrG rArArA rUrCrU rGrGrA rUrArG rGrUrU rUrUrA rGrArG rCrUrA rUrGrC rU/AltR2/ | IDT | N/A | crRNA to make qns-1(ju1563) |
| Sequence-based reagent | crRNA: esyt-2:/AltR1/rGrG rUrUrU rCrArG rUrArA rUrUrG rUrGrG rGrCrU rGrUrU rUrUrA rGrArG rCrUrA rUrGrC rU/AltR2/ | IDT | N/A | crRNA to make esyt-2(ju1409) |
| Sequence-based reagent | crRNA: esyt-2:/AltR1/rGrU rGrCrA rCrUrU rArCrG rGrGrU rUrUrGrU rArGrG rGrUrU rUrUrA rGrArG rCrUrA rUrGrC rU/AltR2/ | IDT | N/A | crRNA to make esyt-2(ju1409) |
| Peptide, recombinant protein | Protein: Cas9-NLS purified protein | QB3 MacroLab, UC Berkley | N/A | |
| Peptide, recombinant protein | Phusion High-Fidelity DNA polymerases | Thermo Scientific | Cat#F530L | |
| Peptide, recombinant protein | DreamTaq DNA polymerases | Thermo Scientific | Cat#EP0705 | |
| Commercial assay or kit | | | | |
| Chemical compound, drug | 5-fluoro-2-deoxy uridine | Sigma-Aldrich | Cat#50-91-9 | |
| Software, algorithm | ImageJ | NIH image | RRID: SCR_003070 | |
| Software, algorithm | ZEN | Zeiss | | https://www.zeiss.com/microscopy/us/downloads/zen.html |
| Software, algorithm | Zeiss LSM Data Server | Zeiss | | https://www.zeiss.com/microscopy/us/downloads/lsm-5-series.html |
| Software, algorithm | GraphPad Prism 5 | GraphPad Software, Inc. | RRID:SCR_002798 | |

## Experimental model

The nematode *Caenorhabditis elegans* was used as the experimental model for this study. All experiments were performed with hermaphrodite animals; males were used only for crosses. Unless otherwise indicated, all experiments were carried out with L4 stage animals. Strains were maintained under standard conditions on Nematode Growth Media (NGM) plates seeded with E. coli OP50 bacteria unless mentioned. Wild type was the N2 Bristol strain (*Brenner, 1974*). New strains were constructed using standard procedures and all genotypes confirmed by PCR or sequencing. Extrachromosomal array transgenic lines were generated as described (*Mello et al., 1991*).

## Laser microsurgery of axons (axotomy)

We cut PLM axons and quantified the length of regrown axons as previously described (*Wu et al., 2007*).

## Axotomy imaging with MicroPoint laser

L4 stage animals were immobilized using 2.5 mM levamisole in M9 buffer on 5% agar pads. Using a MicroPoint laser on an Andor spinning disk confocal unit (CSU-W1) with Leica DMi8 microscope, laser axotomy was performed on the PLM axon ~45 µm away from the cell body. Images were taken immediately before and immediately after axotomy (0.81 s) with iXon ultra 888 EMCCD camera.

## Confocal imaging with Airyscan

L4 stage animals were immobilized using 2.5 mM levamisole in M9 buffer on 5% agar pads. PLM mechanosensory neuron cell bodies were imaged using a Zeiss LSM800 equipped with Airyscan. Z-stack planes were taken at 0.2 µm intervals in both mKate2 and GFP channels using Airyscan.

## CRISPR-Cas9 gene editing

We generated the *nmat-2(ju1512), nmat-1(ju1564), qns-1(ju1563),* and *esyt-2(ju1409)* deletion alleles using co-CRISPR (*Arribere et al., 2014*; *Friedland et al., 2013*). We generated the *nmat-2(ju1514)* point mutation allele using the homology-directed genome editing and single-strand oligodeoxynucleotide repair method (*Paix et al., 2017*).

## FUdR treatment

We transferred worms onto plates containing 50 µg/ml 5-fluoro-2-deoxy uridine (FUdR) immediately after axotomy. No offspring were observed after 2 days, confirming FUDR-induced sterility.

## Quantification and statistical analysis

Statistical analysis was performed using GraphPad Prism 5. Significance was determined using unpaired *t*-tests for two samples, one-way ANOVA followed by multiple comparison tests for multiple samples. $p < 0.05$ (*) was considered statistically significant. *$p < 0.05$; **$p < 0.01$; ***$p < 0.001$. Data are shown as mean ± SEM. 'n' represents the number of animals and is shown in graphs.

## Acknowledgements

We thank our members for valuable discussions and Laura Toy for assistance in strain construction. We thank Dr. S Mitani and National Bioresource Project of Japan, and the *Caenorhabditis* Genetics Center (funded by NIH Office of Research Infrastructure Programs P40 OD010440) for strains. KWK received an American Heart Association postdoctoral fellowship and Hallym University research funds (HRF-201809–014), MGA received a Canadian Institutes of Health Research Postdoctoral Fellowship (MFE-146808), and SJC was a receipient of NIH K99 (NS097638). This work was supported by NIH R01 grants to YJ, and ADC (NS057317 and NS093588).

## Additional information

### Funding

| Funder | Grant reference number | Author |
|---|---|---|
| American Heart Association | 13POST14800057 | Kyung Won Kim |
| Hallym University Research Fund | HRF-201809-014 | Kyung Won Kim |
| Canadian Institutes of Health Research | MFE-146808 | Matthew G Andrusiak |
| National Institutes of Health | NS057317 | Yishi Jin<br>Andrew D Chisholm |
| National Institutes of Health | NS093588 | Yishi Jin<br>Andrew D Chisholm |
| National Institute of Health | K99 NS097638 | Salvatore J Cherra III |

The funders had no role in study design, data collection and interpretation, or the decision to submit the work for publication.

### Author contributions

Kyung Won Kim, Conceptualization, Resources, Data curation, Formal analysis, Validation, Investigation, Visualization, Methodology, Writing—original draft, Writing—review and editing; Ngang Heok Tang, Christopher A Piggott, Matthew G Andrusiak, Seungmee Park, Resources, Data curation, Validation, Investigation, Visualization, Writing—original draft, Writing—review and editing; Ming Zhu, Resources, Data curation, Validation, Investigation, Writing—review and editing; Naina Kurup, Resources, Validation, Investigation, Writing—review and editing; Salvatore J Cherra III, Zilu Wu, Resources, Validation, Investigation; Andrew D Chisholm, Yishi Jin, Conceptualization, Data curation, Supervision, Funding acquisition, Writing—original draft, Project administration, Writing—review and editing

### Author ORCIDs

Kyung Won Kim [iD] http://orcid.org/0000-0002-8252-6203
Andrew D Chisholm [iD] http://orcid.org/0000-0001-5091-0537
Yishi Jin [iD] http://orcid.org/0000-0002-9371-9860

### Decision letter and Author response

Decision letter https://doi.org/10.7554/eLife.39756.021
Author response https://doi.org/10.7554/eLife.39756.022

## Additional files

### Supplementary files

• Transparent reporting form
DOI: https://doi.org/10.7554/eLife.39756.019

### Data availability

All data generated or analysed during this study are included in the manuscript and supporting files.

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
