## [Decision Letter]

Thank you for submitting your article "Expanded genetic screening in *C. elegans* identifies new regulators and an inhibitory role for NAD^+^ in axon regeneration" for consideration by *eLife*. Your article has been reviewed by three peer reviewers, and the evaluation has been overseen by a Reviewing Editor and Eve Marder as the Senior Editor. The following individuals involved in review of your submission have agreed to reveal their identity: Marc Hammarlund (Reviewer #2); Benjamin Podbilewicz (Reviewer #3).

The reviewers have discussed the reviews with one another and the Reviewing Editor has drafted this decision to help you prepare a revised submission.

Summary:

In their manuscript entitled "Expanded genetic screening in *C. elegans* identifies new regulators and an inhibitory role for NAD^+^ in axon regeneration" Kim and colleagues expand on a previous screen also from the Jin lab and examine the role of 611 genes, selected for by genetic homology to humans and mutant availability, in axon regeneration using a single axon laser ablation model. The authors report that 82 of these knockouts significantly altered axon regeneration, 49 of which were important for normal regeneration, and 33 that would normally inhibit regeneration. The identified genes were distributed across a broad set of pathways and molecular functions and the authors do a nice job of covering the relevant background for such a wide set topics.

Their new findings consist of:

- An unexpected inhibitory role for the NAD^+^ salvage pathway (via NAD^+^ itself) in axon regeneration;

- A role for junctophilin 1 in axon fusion;

- Relocalization of extended synaptotagmin-1 to ER-PM contact sites after axotomy;

- Strong functional redundancy in lipid metabolic pathways that can mask regenerative/degenerative effects;

- Inhibition of regeneration by influenza virus NS1A protein largely due to faster initiation of regrowth independent of growth cone formation.

In addition they build on themes from their previous screen by adding candidate regeneration modulators related to the ECM and basement membrane (*emb-9, epi-1, adt-1, adt-3, mig-17*), trafficking (*rab-8*), plasma membrane and resealing (*nex-1, nex-2*), microtubule acetylation (*mec-17, atat-2*) and actin (*fli-1, twf-2*). They also report a growth-potentiating role for the orphan receptor *mfsd-6*.

Overall the data are of good quality, the work of cataloguing so many genes and gene pathways involved in axon regeneration is important, and the presentation of such a daunting dataset is appropriate for such a format. Further, attempts at uncovering gene product interactions and redundancy are to be commended.

The reviewers suggested a number of experiments and rewriting to improve the manuscript. After discussion among the reviewers, two experiments came to the top of the list.

Essential revisions:

1) Subsection 2 The conserved enzyme NMNAT and its product NAD^+^ inhibit axon regeneration”, first and second paragraphs: *nmat-2* mutants are sterile, and have an enhanced PLM regrowth, while *nmat-1* are not fertile and have no changes in regrowth. Is it possible that sterility, or genes affecting fertility affect PLM regrowth? To test this you can add FUDR to plates or use glp mutants. The reviewers felt that this is an experiment that you should be able to carry out.

2) Another experiment that two of the reviewers pointed out is the apparent conflict between the role of the NAD^+^ salvage pathway reported here for PLM axon regeneration and its action in slowing Wallerian degeneration. If NAD^+^ rundown acts as an injury signal, then a lowered concentration could elicit a faster response to injury that is blocked by NAD^+^ supplementation. It would be very helpful if the authors could address the role of NAD^+^ levels in triggering the injury signal, perhaps by a delayed addition of NAD^+^ to the samples instead of incubating it from before the time of injury. If depletion is only needed to trigger the regeneration response, then later supplementation would not affect further growth. A dosage dependent response might also explain this. The reviewers felt that this could be addressed by varying the dosage of NAD^+^ or time of addition. We will leave it to you to decide if this experiment is doable within the time limit of submission, as the reviewers feel it will add, but it is not critical.

---

## [Author Response]

Essential revisions:1) Subsection 2 The conserved enzyme NMNAT and its product NAD^+^ inhibit axon regeneration”, first and second paragraphs: nmat-2 mutants are sterile, and have an enhanced PLM regrowth, while nmat-1 are not fertile and have no changes in regrowth. Is it possible that sterility, or genes affecting fertility affect PLM regrowth? To test this you can add FUDR to plates or use glp mutants. The reviewers felt that this is an experiment that you should be able to carry out.

We tested the axon regeneration effects of wild type and *nmat-1* cultured on FUDR and such treatment caused sterility, but no significant differences in PLM regrowth. The data is presented in revised Figure 3—figure supplement 1. Furthermore, we have previously shown that ablating germline by laser in L1 does not affect PLM axon regeneration in the wild type backgrounds (Kim et al., 2018, Figure 2B, lane 1 and 2). We conclude that germline sterility per se is unlikely to influence PLM regrowth.

Accordingly, we have revised the text:: ‘To address whether sterility of the animals might contribute to the observed effects on axon regrowth, we cultured animals on 5’fluoro-2’ deoxyuridine (FUdR) and found that neither wild type and *nmat-1(0)* grown in FUdR showed increased PLM regrowth (Figure 2—figure supplement 1). Additionally, we have previously reported that sterile animals following germline ablation do not affect PLM regrowth (Kim et al., 2018). Thus, we conclude that NMAT-2’s role in axon regrowth is independent of animal fertility.’

2) Another experiment that two of the reviewers pointed out is the apparent conflict between the role of the NAD^+^ salvage pathway reported here for PLM axon regeneration and its action in slowing Wallerian degeneration. If NAD^+^ rundown acts as an injury signal, then a lowered concentration could elicit a faster response to injury that is blocked by NAD^+^ supplementation. It would be very helpful if the authors could address the role of NAD^+^ levels in triggering the injury signal, perhaps by a delayed addition of NAD^+^ to the samples instead of incubating it from before the time of injury. If depletion is only needed to trigger the regeneration response, then later supplementation would not affect further growth. A dosage dependent response might also explain this. The reviewers felt that this could be addressed by varying the dosage of NAD^+^ or time of addition. We will leave it to you to decide if this experiment is doable within the time limit of submission, as the reviewers feel it will add, but it is not critical.

We agree with the reviewers’ opinions, as well as observations from other investigators on *Drosophila* and mammalian neurons that NAD^+^ effects in axon regeneration or degeneration are highly dose-dependent. In the course of revision, we tested acute treatment with NAD^+^ (i.e., a delayed addition of NAD^+^ right after axotomy), and observed no major effects on PLM regrowth. In the course of this experiment, however, we observed that NAD^+^ supplementation is extremely sensitive to minor variations in how the NAD^+^ is prepared. We suspect that this might reflect the labile nature of NAD^+^ and the variability in delivery via the food supply *C. elegans*. In view of these inconsistencies in the effects of supplementation, which will take additional time to resolve, we have removed the NAD^+^ supplementation data.